# THEORETICAL GUARANTEES FOR HIGH ORDER TRAJECTORY REFINEMENT IN GENERATIVE FLOWS

## ABSTRACT

Flow matching has emerged as a powerful framework for generative modeling, offering computational advantages over diffusion models by leveraging deterministic Ordinary Differential Equations (ODEs) instead of stochastic dynamics. While prior work established the worst case optimality of standard flow matching under Wasserstein distances, the theoretical guarantees for higher-order flow matching - which incorporates acceleration terms to refine sample trajectories - remain unexplored. In this paper, we bridge this gap by proving that higher-order flow matching preserves worst case optimality as a distribution estimator. We derive upper bounds on the estimation error for second-order flow matching, demonstrating that the convergence rates depend polynomially on the smoothness of the target distribution (quantified via Besov spaces) and key parameters of the ODE dynamics. Our analysis employs neural network approximations with carefully controlled depth, width, and sparsity to bound acceleration errors across both small and large time intervals, ultimately unifying these results into a general worst case optimal bound for all time steps.

## 1 INTRODUCTION

Machine learning has been profoundly transformed through generative models' capacity to produce authentic and varied outputs in multiple application areas. Notably, techniques such as diffusion models (Ho et al., 2020), Generative Adversarial Networks (GANs) (Goodfellow et al., 2014), and flow matching approaches (Lipman et al., 2023; Liu et al., 2023) have become essential instruments for generating and augmenting datasets. These frameworks employ advanced structural designs to approximate intricate statistical distributions, converting unstructured noise into semantically rich artifacts. Contemporary systems, like text-guided image generators, map linguistic prompts to vivid digital imagery or photorealistic scenes (Zhang et al., 2023a), whereas cutting-edge text-to-video architectures synthesize temporally coherent multimedia sequences (Ho et al., 2022). The discrete flow matching paradigm (Gat et al., 2024) adapts continuous flow methodologies to categorical spaces through meticulous distribution synchronization via adaptable mappings, extending flow-driven generation's utility to combinatorial domains like natural language processing and algorithmic code synthesis. The progressive refinement of these approaches highlights their expanding impact in AI development, as generative systems increasingly master nuanced data patterns while delivering superior synthetic results.

While both diffusion models and flow matching have shown impressive practical results, their theoretical properties - particularly their statistical efficiency as distribution estimators - have remained less understood. Recent theoretical studies (Oko et al., 2023) have established that diffusion models achieve worst case optimal estimation rates under specific function space assumptions, positioning them as theoretically sound generative frameworks. Specifically, it has been shown that diffusion models attain nearly worst case optimal convergence rates in total variation and Wasserstein distances when the true density lies in the Besov space. Flow matching, a recently proposed alternative, simplifies the generative process by replacing stochastic diffusion dynamics with a deterministic ODE formulation. This approach offers several computational advantages, such as bypassing the need for iterative stochastic sampling, while still preserving the ability to interpolate between distributions. Prior work (Fukumizu et al., 2025) suggests that flow-matching methods can achieve nearly worst case optimal convergence rates under the $p$-Wasserstein distance for $(1 \leq p \leq 2)$, making them competitive with diffusion models from a statistical perspective.

One critical aspect of generative modeling is the role of higher-order matching (Chen et al., 2025a) in improving efficiency and accuracy. In diffusion models, high-order solvers and shortcut methods (Frans et al., 2025) enable rapid sampling with minimal degradation in generation quality. Similarly, in flow matching, incorporating higher-order terms into the ODE formulation could further refine the trajectory of sample evolution, particularly for one-step shortcut generation. However, a rigorous understanding of the impact of high-order corrections on worst case optimality remains an open problem.

In this work, we extend the theoretical understanding of flow-matching models by investigating the worst case optimality of higher-order matching.

Our key contribution is that we establish that high-order flow matching remains a worst case optimal distribution estimator, reinforcing its theoretical guarantees. We derive upper bounds on the estimation error for higher-order flow-matching methods, revealing how key parameters influence convergence (Theorem 4.1 and Theorem 4.2).

Our findings not only bridge the theoretical gap between diffusion models and higher-order flow-matching approaches but also provide insights for improving generative model efficiency. By demonstrating that high-order flow matching preserves worst case optimality, we offer a solid foundation for further exploration of fast and reliable generative techniques.

**Roadmap.** In Section 2, we review relevant literature related to our study. Section 3 provides the necessary background. In Section 4, we present our main results on bounded worst case error. Section 5 details the methodology used to obtain these results. Finally, in Section 6, we conclude the paper.

## 2 RELATED WORK

**Diffusion Models.** Diffusion models have acquired substantial recognition due to their ability to produce high-fidelity images by iteratively enhancing noisy inputs, illustrated by DiT (Peebles & Xie, 2023) and U-ViT (Bao et al., 2023). In general, such methods employ a forward mechanism that incrementally infuses noise into a pristine image and a corresponding backward procedure that methodically eliminates noise, thus probabilistically reconstructing the underlying data distribution. Early investigations (Song & Ermon, 2019; Song et al., 2020) established the theoretical bearings of this denoising paradigm, introducing score-matching and continuous-time diffusion frameworks that significantly enhanced the variety and quality of generated samples. Subsequent research explored more efficient training and sampling protocols (Lu et al., 2022; Shen et al., 2025a;b), aiming to reduce computational expense and expedite convergence without compromising image fidelity. Additional work capitalizes on latent representation learning to create compressed embeddings, thereby streamlining both training and inference (Rombach et al., 2022; Hu et al., 2024f). Such latent-based strategies align smoothly with modern neural designs and readily generalize to various modalities, underscoring the flexibility of diffusion processes in capturing intricate data distributions. Concurrently, recent efforts have examined multi-scale noise scheduling and adaptive step-size tactics to bolster convergence reliability while preserving high-resolution details in generated outputs (Lovelace et al., 2024; Feng et al., 2024a; Rout et al., 2024; Jiang et al., 2025; Luo et al., 2024). Furthermore, a multitude of studies have served as supplementary inspirations for our work (Xu et al., 2022a; Dax et al., 2023; Pooladian et al., 2023; Wang et al., 2023c;a; Shen et al., 2025a;b; Wang et al., 2024b; Chen & Lipman, 2024; Klein et al., 2024; Chen et al., 2025c; Cao et al., 2025; Cheng et al., 2024; Wang et al., 2023b; Feng et al., 2024b; Liu et al., 2024; Hu et al., 2024e).

**Flow Matching.** Flow matching is a generative modeling technique that trains continuous normalizing flows by learning a vector field to transform a noise distribution into the data distribution, avoiding full path simulation (Lipman et al., 2023; Liu et al., 2023; Albergo & Vanden-Eijnden, 2023; Heitz et al., 2023). This approach, rooted in efficient CNF training, has spurred a range of recent advancements. (Benton et al., 2024) provide theoretical error bounds for flow-matching methods, ensuring precision under deterministic sampling across varied datasets. (Isobe et al., 2024) extend flow matching to conditional generation using a generalized continuity equation, enabling applications like style transfer and controlled synthesis. (Kerrigan et al., 2024) generalize flow match-

ing to function spaces, supporting the generation of infinite-dimensional data such as sequences or trajectories. (Klein et al., 2024) introduce equivariant flow matching, harnessing physical symmetries for efficient training on symmetric data like molecular structures or particle systems. (Zhang et al., 2024b) explore the minimax optimality of score-based diffusion models, offering insights for flow matching when paired with diffusion paths. (Song et al., 2021) introduce probability flow ODEs, enhancing flow matching with faster training via Gaussian diffusion structures. (Rezende & Mohamed, 2015) establish the normalizing flow framework, providing a foundation for flow matching's vector field approach. (Vincent, 2011) propose denoising score matching, which can be used to accelerate flow matching's sampling efficiency. (Grathwohl et al., 2019) develop free-form CNFs, inspiring flow matching's simulation-free design. (Papamakarios et al., 2021) review normalizing flows, contextualizing flow matching's advancements in image generation. (Albergo & Vanden-Eijnden, 2023) use stochastic interpolants in flow matching, improving its flexibility for finite-dimensional spaces. (Chen & Lipman, 2024) adapt flow matching to Riemannian manifolds, broadening its geometric applications.

**High-order ODE Gradient in Diffusion Models.** Higher-order gradient-based techniques, such as TTMs (Kloeden & Platen, 1992), extend well beyond DDMs in their range of applicability, systematically incorporating underlying principles that guide specialized numerical methods for SDEs across disciplines. The solver proposed in (Djeumou et al., 2022) relies on higher-order derivatives via a fixed-order Taylor expansion for numerical integration, providing significant speedups in training neural ODEs. The regularization method in (Kelly et al., 2020) incorporates higher-order derivatives of solution trajectories through Taylor-mode automatic differentiation, encouraging learned dynamics that are easier to solve. The regularization approach in (Finlay et al., 2020) relies on higher-order derivatives to enforce simpler neural ODE dynamics with optimal transport and stability constraints, reducing solver steps and accelerating training. Neural ODEs in (Chen et al., 2018) define continuous-depth models by parameterizing the hidden state's derivative with a neural network, enabling constant memory cost and scalable backpropagation through higher-order derivatives. Neural ODEs in (Grathwohl et al., 2018) yield continuous-time invertible generative models, estimating log-density via a Jacobian trace that leverages higher-order derivatives. Moreover, outside the realm of machine learning, extensive research on higher-order TTMs has been devoted to addressing both stiff (Chang & Corliss, 1994) and non-stiff (Corliss & Chang, 1982; Chang & Corliss, 1994) systems.

## 3 PRELIMINARY

In this section, we introduce the preliminaries. Section 3.1 presents our notation. Section 3.2 defines the neural network class. Section 3.3 covers core flow matching concepts, including the divergence, pushforward operator, generative vector fields, transport and continuity equations, and a bound on the vector field's magnitude.

### 3.1 NOTATIONS

We use $I^d$ to denote the $d$-dimensional cube $[-1, 1]^d$. We use $A \setminus B$ to denote the set difference of two sets $A$ and $B$, i.e., $A \setminus B := \{ x \in A : x \notin B \}$. For any positive integer $n$, we use $[n]$ to denote set $\{1, 2, \cdots, n\}$. We use $\mathbb{E}[\cdot]$ to denote the expectation. For two vectors $x \in \mathbb{R}^n$ and $y \in \mathbb{R}^n$, we use $\langle x, y \rangle$ to denote the inner product between $x, y$, i.e., $\langle x, y \rangle = \sum_{i=1}^n x_i y_i$. We use $\mathbf{1}[C]$ to represent an indicator variable that takes the value 1 when condition $C$ holds and 0 otherwise. We use $x_{i,j}$ to denote the $j$-th coordinate of $x_i \in \mathbb{R}^n$. We use $\circ$ to denote the composition of the function. For $k > n$, for any matrix $A \in \mathbb{R}^{k \times n}$, we use $\|A\|$ to denote the spectral norm of $A$, i.e. $\|A\| := \sup_{x \in \mathbb{R}^n} \|Ax\|_2 / \|x\|_2$. We use $\|x\|_p$ to denote the $\ell_p$ norm of a vector $x \in \mathbb{R}^n$, i.e. $\|x\|_1 := \sum_{i=1}^n |x_i|$, $\|x\|_2 := (\sum_{i=1}^n x_i^2)^{1/2}$. We use $\det(A)$ to denote the determinant of matrix $A$. Let $I_d \in \mathbb{R}^{d \times d}$ denote an identity matrix. We use $\mathcal{N}_d$ to denote a $d$-dimensional multivariate normal (Gaussian) distribution. We use $n!!$ to denote the double factorial of $n \in \mathbb{N}$, defined as $n!! := n \times (n - 2) \times \cdots \times 2$ if $n$ is even, and $n!! := n \times (n - 2) \times \cdots \times 1$ if $n$ is odd. We define the $L^2$ norm over region $S$ as $\|f\|_{L^2(S)} := (\int_S |f(x)|^2 \mathrm{d}x)^{1/2}$. In addition to $O(\cdot)$ notation, for two functions $f, g$, we use the shorthand $f \lesssim g$ (resp. $\gtrsim$) to indicate that $f \leq Cg$ (resp. $\geq$) for

an absolute constant $C$. For two vectors $x \in \mathbb{R}^n$ and $y \in \mathbb{R}^m$, we use $[x, y] \in \mathbb{R}^{n+m}$ to denote the concatenation of $x$ and $y$. We use $*$ to denote convolution.

## 3.2 Neural Network Class

To precisely characterize the networks we analyze, we introduce the following definition, parameterized by height $L$, width $W$, sparsity $S$, and norm bound $B$.

**Definition 3.1** (Neural Network Class, Definition 2.1 on Page 3 in (Oko et al., 2023))**.** *Let $A^{(i)} \in \mathbb{R}^{W_i \times W_{i+1}}, b^{(i)} \in \mathbb{R}^{W_{i+1}}$. Let $W := (W_1, W_2, \ldots, W_L)$. Let $\sum_{i=1}^{L}(\|A^{(i)}\|_0 + \|b^{(i)}\|_0) \leq S$. Let $\max_{i \in [L]} \max\{\|A^{(i)}\|_\infty, \|b^{(i)}\|_\infty\} \leq B$. We define class of neural networks $\Phi(L, W, S, B)$ with height $L$, width $W$, sparsity constraint $S$, and norm constraint $B$ as $\Phi(L, W, S, B) := \{(A^{(L)}\mathrm{ReLU}(\cdot) + b^{(L)}) \circ \cdots \circ (A^{(1)}x + b^{(1)})\}$.*

## 3.3 Basic Concept of Flow Matching

We now define the divergence of a vector-valued function, a fundamental concept in vector calculus.

**Definition 3.2** (Divergence)**.** *Let $f : \mathbb{R}^n \to \mathbb{R}^n$, $f(x) = (f_1(x), f_2(x), \ldots, f_n(x))$. We define the divergence of $f(x)$ as: $\nabla \cdot f(x) := \sum_{i=1}^{n} \frac{\mathrm{d}f_i(x)}{\mathrm{d}x_i}$ This is obvious to see $\nabla \cdot f(x) = \langle \frac{\mathrm{d}f(x)}{\mathrm{d}x}, \mathbf{1}_n \rangle$.*

**Fact 3.3** (Derivative of Determinant)**.** *For a matrix $X \in \mathbb{R}^{n \times m}$, we have $\frac{\mathrm{d}}{\mathrm{d}t} \det(X) = \det(X) \cdot \mathrm{tr}[X^{-1}\frac{\mathrm{d}X}{\mathrm{d}t}]$*

The conditional probability distribution $P_t(x_{1,t} \mid y)$ is defined as follows

**Definition 3.4** (Gaussian Conditional Distribution)**.** *We define the distribution $P_t(x_{1,t} \mid y)$ as: $P_t(x_{1,t} \mid y) = \mathcal{N}_d(x_{1,t}; \beta_t y, \alpha_t^2 I_d)$ where $x_{1,t}$ is the trajectory defined in Definition 3.10, $\mathcal{N}_d$ is a $d$-dimensional multivariate Gaussian distribution.*

We formally define the pushforward operator, a key concept for transforming probability densities.

**Definition 3.5** (Pushforward Operator, Implicit in Section 2 of (Lipman et al., 2023))**.** *Given a continuously differentiable vector field $v_t$ and a probability density $p_0$, the pushforward operator $*$ is defined as $[v_t]_* p_0 := p_0(v_t^{-1}(x)) \cdot \det(\frac{\mathrm{d}v_t^{-1}}{\mathrm{d}x}(x))$ where $v_t^{-1}$ denote the inverse function of $v_t$, i.e., $v_t^{-1} \circ v_t$ is identity mapping.*

Leveraging the pushforward operator, we define the condition for a vector field to generate a specific probability density.

**Definition 3.6** ($v_t(x)$ Generate $p_t(x)$)**.** *Given a continuously differentiable vector field $v_t$ and a probability density $p_0$, we say that $v_t$ generate $p_t$ if $p_t = [v_t]_* p_0$.*

We now present a key result connecting the generative vector field with the evolution of the conditional probability density over time.

**Lemma 3.7** (Transport Equation with the Conditional Vector Field, (Lipman et al., 2023))**.** *If $v_t(x_1)$ generates $p_t(x_1)$, then the following transport equation with the conditional vector field holds:*

$$\frac{\mathrm{d}p_t(x_1 \mid y)}{\mathrm{d}t} = -\nabla \cdot (v_t(x_1 \mid y)p_t(x_1 \mid y)).$$

We define the vector field $v_t$ as the conditional expectation of $v_t(x_1 \mid y)$ with respect to the conditional distribution $p_{1 \mid t}$.

**Definition 3.8.** *The vector field $v_t$ is $v_t(x_1) = \mathbb{E}_{y \sim p_{1 \mid t}}[v_t(x_1 \mid y)] = \int v_t(x_1 \mid y) \cdot \frac{p_t(x_1 \mid y)q_1(y)}{p_t(x_1)} \mathrm{d}y.$*

Under the assumptions of the transport equation and the definition of the vector field $v_t$, we derive the first-order continuity equation.

**Lemma 3.9** (First-order Continuity Equation, (Lipman et al., 2023))**.** *The continuity equation is given as*

$$\frac{\mathrm{d}p_t(x_1)}{\mathrm{d}t} = -\nabla \cdot (v_t(x_1)p_t(x_1)).$$

We present the general framework of flow matching and its high-order rectification as follows.

**Definition 3.10.** *We define the vector field $x_t$ as follows: $x_{1,t} := a_t x_0 + b_t x_1$, where $\alpha_t$ and $\beta_t$ are functions related to $t$, $x_{1,0}$ and $x_{1,1}$ are initial distribution and target distribution respectively. Based on this, we can give the first order rectification of $x_t$ as: $x'_{1,t} = \alpha'_t x_{1,0} + \beta'_t x_{1,1}$. We can give the first order rectification of $x_t$ as: $x''_{1,t} = \alpha''_t x_{1,0} + \beta''_t x_{1,1}$.*

## 4 MAIN RESULT

In this section, we present our main theoretical results on bounding the acceleration error in second-order flow matching using neural network approximations. In Section 4.1, we establish a bound for small values of $t$. Section 4.2 provides a complementary bound for large values of $t$.

### 4.1 BOUNDS ON SECOND ORDER FLOW MATCHING SMALL $t$

We now present our main theorem, which bounds the acceleration error for sufficiently small $t$ using a neural network approximation.

**Theorem 4.1** (Main Theorem, Bound Acceleration Error under Small $t$, Informal Version of Theorem D.1)**.** *If the following conditions hold:*

- *Assume Assumption 5.2, 5.3, 5.4, 5.6, 5.8, 5.9 hold.*

- *Let $C_6$ be a constant independent of $t$.*

- *Let $x_1$ be the trajectory, $x_2 := \phi_1(x_1, t)$ where $\phi_1$ is the neural network in Lemma B.3.*

- *Let $x$ be defined as the concatenation of $x_1$ and $x_2$, i.e., $x := [x_1, x_2]$.*

*Then there is a neural network $u_1 \in \mathcal{M}(L, W, S, B)$ and a constant $C$, which is independent of $t$, such that, for sufficiently large $N$,*

$$\int \|u_1(x, t) - a_t(x_1)\|_2^2 \cdot p_t(x_1) \mathrm{d}x_1 \leq C_6 \cdot (\alpha_t''^2 \log N + \beta_t''^2) \cdot N^{-\frac{2s}{d}},$$

*for any $t \in [T_0, 3T_*]$, where*

$$L = O(\log^4 N), \|W\|_\infty = O(N \log^6 N),$$
$$S = O(N \log^8 N), B = \exp(O((\log N) \cdot (\log \log N))).$$

### 4.2 BOUNDS ON SECOND ORDER FLOW MATCHING LARGE $t$

We present a complementary result to Theorem 4.1, providing a bound on the acceleration error for large $t$.

**Theorem 4.2** (Main Theorem, Bound Acceleration Error under Large $t$, Informal Version of Theorem E.1)**.** *If the following conditions hold:*

- *Fix $t_* \in [T_*, 1]$ and take arbitrary $\eta > 0$.*

- *Assume Assumption 5.2, 5.3, 5.4, 5.6, 5.8, 5.9 hold.*

- *Let $C_7$ be a constant independent of $t$.*

- *Let $x_1$ be the trajectory, $x_2 := \phi_2(x_1, t)$ where $\phi_2$ is the neural network in Lemma B.4.*

- *Let $x$ be defined as the concatenation of $x_1$ and $x_2$, i.e., $x := [x_1, x_2]$.*

*Then there is a neural network $u_2 \in \mathcal{M}(L, W, S, B)$ and a constant $C$, which is independent of $t$, such that, for sufficiently large $N$,*

$$\int \|u_2(x, t) - a_t(x_1)\|_2^2 \cdot p_t(x_1) \mathrm{d}x_1 \leq C_7 \cdot ((\alpha_t'')^2 \log N + (\beta_t'')^2) \cdot N^{-\eta}$$

*for any $t \in [2t_*, 1]$, where*

$$L = O(\log^4 N), \|W\|_\infty = O(N \log^6 N),$$
$$S = O(N \log^8 N), B = \exp(O((\log N) \cdot (\log \log N))).$$

## 5 TECHNIQUE OVERVIEW

In this section, we introduce the key technical tools, assumptions, and supporting lemmas that underpin our analysis. In Section 5.1, we introduce the second-order continuity equation and the concept of vector fields generating a probability density in this context. Section 5.2 details the core assumptions on the target probability distribution and related parameters. Section 5.3 derives crucial bounds on the acceleration term. Section 5.4 presents a lemma for approximating the initial probability density. Finally, Section 5.5 establishes the existence and properties of specialized neural network architectures, i.e., sub-networks, that perform fundamental operations, and Section 5.6 introduces the function composition theorem and definition.

### 5.1 SECOND ORDER CONTINUITY

We introduce the second-order continuity equation, which describes the evolution of the second derivative of the probability density with respect to time.

**Lemma 5.1** (Second-order Continuity Equation). *Let $x_1$ be the trajectory, $v_t$ be the velocity, $a_t$ be the acceleration, and $p_t$ be the density. The second-order continuity equation is given as*

$$\frac{\mathrm{d}^2 p_t(x_1)}{\mathrm{d}t^2} = -\nabla \cdot \left( v_t(x_1) \cdot \frac{\mathrm{d}p_t(x_1)}{\mathrm{d}t} + a_t(x_1) \cdot p_t(x_1) \right).$$

### 5.2 BASIC ASSUMPTIONS

We begin by establishing the regularity assumptions on the target probability distribution $P_0$.

**Assumption 5.2.** *We use $I_N^d$ to represent the contracted cube $[-1 + N^{-(1-\kappa\delta)}, 1 - N^{-(1-\kappa\delta)}]^d$, where $N$ relates to sample size and $\kappa, \delta$, which satisfy Assumption 5.4. The target probability $P_0$ has support $I^d$ and its density $p_0$ satisfies $p_0 \in B_{p',q'}^s(I^d)$ and $p_0 \in B_{p',q'}^{\check{s}}(I^d \backslash I_N^d)$ with $\check{s} > \max\{6s, 1\}$.*

We assume that $p_0(x_1)$ is bounded as follows:

**Assumption 5.3.** *There exists constant $C_0 > 0$ such that for density $p_0$ $C_0^{-1} \le p_0(x_1) \le C_0$, $\forall x_1 \in I^d$*

**Assumption 5.4.** *For $\kappa \ge 1/2$, $b_0 > 0$, $\widetilde{\kappa} > 0$, and $\widetilde{b}_0 > 0$, we assume that for sufficiently small $t \ge T_0$ $\alpha_t = b_0 t^\kappa$, $1 - \beta_t = \widetilde{b}_0 t^{\widetilde{\kappa}}$ Also, there is $D_0 > 0$ such that $D_0^{-1} \le \alpha_t^2 + \beta_t^2 \le D_0$, $\forall t \in [T_0, 1]$.*

**Assumption 5.5.** *For the first-order derivative of $\alpha_t$ and $\beta_t$, i.e., $\alpha_t'$ and $\beta_t'$, we assume there is a constant $K_0 > 0$ that $|\alpha_t'| + |\beta_t'| \le N^{K_0}$, $\forall t \in [T_0, 1]$.*

**Assumption 5.6.** *For the second-order derivative of $\alpha_t$ and $\beta_t$, i.e., $\alpha_t''$ and $\beta_t''$, we assume there is a constant $K_0 > 0$ that $|\alpha_t''| + |\beta_t''| \le N^{K_0}$, $\forall t \in [T_0, 1]$.*

**Assumption 5.7.** *Let $s$ be a Besov space constant in Definition B.2. Let $\kappa$ satisfies Assumption 5.4. Let $T_0$ be defined in Definition 5.10. Let $R_0$ be a constant fixed as $R_0 \ge \frac{s+1}{\min\{\kappa, \bar{\kappa}\}}$. If $\kappa = 1/2$, then there exist constants $b_1 > 0$ and $D_1 > 0$ such that for any $0 \le \gamma < R_0$, $\int_{T_0}^{N^{-\gamma}} (\alpha_t'^2 + \beta_t'^2) \mathrm{d}t \le D_1 \cdot \log^{b_1} N$. holds for the second-order derivative of $\alpha_t$ and $\beta_t$, i.e., $\alpha_t''$ and $\beta_t''$,*

**Assumption 5.8.** *Let $s$ be a Besov space constant in Definition B.2. Let $\kappa$ satisfies Assumption 5.4. Let $T_0$ be defined in Definition 5.10. Let $R_0$ be a constant fixed as $R_0 \ge \frac{s+1}{\min\{\kappa, \bar{\kappa}\}}$. If $\kappa = 1/2$, then there exist constants $b_1 > 0$ and $D_1 > 0$ such that for any $0 \le \gamma < R_0$, $\int_{T_0}^{N^{-\gamma}} (\alpha_t''^2 + \beta_t''^2) \mathrm{d}t \le D_1 \log^{b_1} N$. holds for the second-order derivative of $\alpha_t$ and $\beta_t$, i.e., $\alpha_t''$ and $\beta_t''$,*

**Assumption 5.9.** *There is a constant $C_L > 0$ such that $\|\frac{\mathrm{d}}{\mathrm{d}x_1} \int y p_t(y \mid x_1) \mathrm{d}y\| \le C_L$ for any $t \in [T_0, 1]$.*

## 5.3 BOUNDS ON ACCELERATION

**Definition 5.10** (Time Variables and Partition). *We define the key time-related variables as follows:*

- *We define the initial time $T_0$ as $T_0 := N^{-R_0}$*

- *We define $T_*$ as $T_* := N^{-(\kappa^{-1}-\delta)/d}$.*

- *We define $t_{j_*} \in [T_*, 3T_*]$ as a boundary time where different error bounds are applied for generalization analysis.*

- *For $j \in [K]$, we define $t_j$ as $t_j := 2t_{j-1}$, specially we define $t_0 := T_0$ and $t_K := 1$.*

We now derive a bound on the magnitude of the acceleration term $a_t(x_1)$.

**Theorem 5.11** (Bound of $a_t$, informal version of Theorem C.1). *Let $x_1$, $\alpha$, $\beta$ be defined in Definition 3.10. Let $C_3 > 0$ be a constant depend on $d$ and $C_0$. Then we can show that $\|a_t(x_1)\|_2 \leq C_3 \cdot (\alpha_t'' \cdot \max\{(\|x_1\|_\infty - \beta_t)/\alpha_t, 1\} + |\beta_t''|)$ for any $x_1 \in \mathbb{R}^d$ and $t \in [T_0, 1]$.*

We derive a bound on the integral of the squared $L_2$ norm of the difference between the acceleration term and its neural network approximation, restricted to a specific domain.

**Lemma 5.12** (Bound $a_t$ with Constant, informal version of Lemma C.2). *Let $\epsilon \in (0, 0.1)$ be a small number. Let $\alpha_t$ and $\beta_t$ be defined in Definition 3.10. For any $C_4 > 0$, we have*

$$\|a_t(x_1)\|_2 \leq C_4 \cdot (\alpha_t'' \sqrt{\log(1/\epsilon)} + |\beta_t''|)$$

*for any $x_1$ with $\|x_1\|_\infty \leq \beta_t + C_4 \alpha_t \sqrt{\log(1/\epsilon)}$ and $t \in [T_0, 1]$.*

This lemma allows us to bound an integral involving the difference between the acceleration and a neural network approximation, under certain conditions.

**Lemma 5.13** (Omit Small Term of Integral on Acceleration, informal version of Lemma C.3). *Let $\omega > 0$ be an arbitrary positive number. Let $u$ be a neural network. Let $C' > 0$ be a constant. Let $D := \{x_1 \in \mathbb{R}^d \mid \|x_1\|_\infty \leq \beta_t + C_4 \alpha_t \sqrt{\log N}\}$ Let $x_1$ be the trajectory, $x_2 := \phi_2(x_1, t)$ where $\phi_2$ is the neural network in Lemma B.4. Let $x$ be defined as the concatenation of $x_1$ and $x_2$, i.e., $x := [x_1, x_2]$. Let $\alpha_t$ and $\beta_t$ be defined in Definition 3.10. Then we have*

$$\int_D \mathbf{1}[p_t(x_1) \leq N^{-\frac{2s+\omega}{d}}] \cdot \|a_t(x_1) - u(x,t)\|_2^2 \cdot p_t(x_1)\mathrm{d}x_1 = o(C' \cdot (\alpha_t''^2 \log N + \beta_t''^2)N^{-\frac{2s}{d}})$$

Before introduce the next bound, we present the bound of Gamma function:

**Lemma 5.14** (Bound Gamma Function, informal version of Lemma C.4). *Let $\ell \in \mathbb{N}$. Let $\psi_\ell(z) := \int_z^\infty r^\ell \exp(-\frac{r^2}{2})\mathrm{d}r$. Then we have the bound that: $\psi_\ell(z) \leq \ell!! \cdot z^{\ell-1} \cdot \exp(-\frac{z^2}{2})$*

We introduce an assumption regarding the boundedness of certain integrals involving the probability density and the vector field.

**Lemma 5.15** (Bounded Integral $a_t$, informal version of Lemma C.5). *Let $\alpha_t$ and $\beta_t$ be defined in Definition 3.10. For any $C_5 > 0$, there is $\widetilde{C} > 0$ such that*

$$\int_{\|x_1\|_\infty \geq \beta_t + C_5 \alpha_t \sqrt{\log(1/\epsilon)}} p_t(x_1) \cdot \|a_t(x_1)\|_2^2 \mathrm{d}x_1 \leq \widetilde{C} \cdot \epsilon^{\frac{C_5^2}{2}} \cdot (\alpha_t''^2 \log^{\frac{d}{2}}(1/\epsilon) + \beta_t''^2 \log^{\frac{d-2}{2}}(1/\epsilon))$$

*hold for any $\epsilon > 0$ and $t \in [T_0, 1]$.*

## 5.4 LINEAR COMBINATION OF FUNCTION

We introduce a lemma concerning the approximation of the initial probability density $p_0$ by a function $f_N$ with a specific form.

**Lemma 5.16** (Lemma B.4 in (Oko et al., 2023)). *Let $s$ satisfy Assumption 5.2, let $\kappa$ satisfy Assumption 5.4. Let $\delta \in (0, 0.1)$. Let $M_{k,j}^d(x)$ be defined in Definition 5.23. Let $i \in [N]$, $m \in [d]$. Let each*

*entry in $j_i$ is bounded, i.e., $-2^{(k_i)_m} - \ell \le (j_i)_m \le 2^{(k_i)_m}$. Let $\mathcal{A}_i \in \mathbb{N}$. There exists a function $f_N$ which is defined as*

$$f_N(x_1) := \sum_{i=1}^{N} \mathcal{A}_i \cdot \mathbf{1}[\|x_1\|_\infty \le 1] \cdot M_{k_i,j_i}^d(x_1) + \sum_{i=N+1}^{3N} \mathcal{A}_i \cdot \mathbf{1}[\|x_1\|_\infty$$

$$\le 1 - N^{-\frac{\kappa^{-1}-\delta}{d}}] \cdot M_{k_i,j_i}^d(x_1),$$

*that satisfies $\|p_0 - f_N\|_{L^2(I_d)} \le C_a N^{-s/d}$, $\|p_0 - f_N\|_{L^2(I_d \setminus I_d^N)} \le C_a N^{-\tilde{s}/d}$, for some $C_a > 0$. It also satisfied $f_N(x_1) = 0$ for any $x_1$ with $\|x_1\|_\infty \ge 1$.*

## 5.5 SUB-NEURAL NETWORK

In this section, we introduce a neural network class capable of achieving certain functionalities.

We first introduce a lemma demonstrating the existence of a neural network that implements a clipping

**Lemma 5.17** (Lemma 20 in (Fukumizu et al., 2025)). *For any $a, b \in \mathbb{R}^d$ with $a_i \le b_i$ for $i \in [d]$, there exists a neural network $\mathsf{clip}(x; a, b) \in \mathcal{M}(L, W, S, B)$ with $L = 2$, $W = (d, 2d, d)^\top$, $S = 7d$, and $B = \max_{1 \le i \le d} \max\{|a_i|, b_i\}$ such that $\mathsf{clip}(x; a, b)_i = \min\{b_i, \max\{x_i, a_i\}\}$ $(i = 1, 2, \ldots, d)$*

*holds. When $a_i = c_a$ and $b_i = c_b$ hold for all $i \in [d]$ and constant $c_a$, $c_b \in \mathbb{R}$, the notation can be simplified as $\mathsf{clip}(x; c_a, c_b) := \mathsf{clip}(x; a, b)$.*

We then present a lemma establishing the existence of a neural network that approximates the reciprocal function within a specified range and with a defined error bound.

**Lemma 5.18** (Lemma 21 in (Fukumizu et al., 2025)). *For any $\epsilon \in (0, 0.1)$, there is $\mathsf{recip}(x') \in \mathcal{M}(L, W, S, B)$ such that $|\mathsf{recip}(x') - \frac{1}{x}| \le \epsilon + \frac{|x-x'|}{\epsilon^2}$ holds for any $x \in [\epsilon, \epsilon^{-1}]$ and $x' \in \mathbb{R}$ with $L = O(\log^2(1/\epsilon))$, $\|W\|_\infty = O(\log^3(1/\epsilon))$, $S = O(\log^4(1/\epsilon))$, and $B = O(\epsilon^{-2})$.*

We present a lemma showing the existence of a neural network that approximates the product of multiple inputs, with a specified error bound.

**Lemma 5.19** (Lemma 22 in (Fukumizu et al., 2025)). *Let $d \ge 2$, $C \ge 1$, $\epsilon_{\mathrm{err}} \in (0, 0.1)$. For any $\epsilon \in (0, 0.1)$, there exists a neural network $\mathsf{mult}(x_1, x_2, \ldots, x_d) \in \mathcal{M}(L, W, S, B)$ with $L = O(d\log(C/\epsilon))$, $\|W\|_\infty = 48d$, $S = O(d\log(C/\epsilon))$, $B = C^d$ such that $|\mathsf{mult}(x'_1, \ldots, x'_d) - \prod_{i=1}^{d} x'_i| \le \epsilon + dC^{d-1}\epsilon_{\mathrm{err}}$, holds for all $x \in [-C, C]^d$ and $x' \in \mathbb{R}^d$ with $\|x - x'\|_\infty \le \epsilon_{\mathrm{err}}$. Also, for all $x \in \mathbb{R}^d$, we can show $|\mathsf{mult}(x)| \le C^d$. If at least one of $x'_i$ is 0, there is $\mathsf{mult}(x'_1, \ldots, x'_d) = 0$. Also, let $\prod_{i=1}^{I} x_{\alpha_i}$ with $\alpha_i \in \mathbb{Z}_+$ and $\sum_{i=1}^{I} \alpha_i = d$, there exists a neural network satisfying that $\mathsf{mult}(x; \alpha) \le \epsilon + dC^{d-1}\epsilon_{\mathrm{err}}$.*

We present a lemma that provides upper and lower bounds for the probability density $p_t(x_1)$ in terms of exponential functions.

**Lemma 5.20** (Lemma 17 in (Fukumizu et al., 2025), Bound of $p_t(x_1)$). *There exists $C_1 > 0$ depend on $d, C_0$ such that*

$$C_1^{-1} \cdot \exp(-\alpha_t^{-2} \cdot \max\{\|x_1\|_\infty - \beta_t, 0\}^2) \le p_t(x_1)$$
$$\le C_1 \cdot \exp(-0.5\alpha_t^{-2} \cdot \max\{\|x_1\|_\infty - \beta_t, 0\}^2)$$

*for any $x_1 \in \mathbb{R}^d$ and $t \in [T_0, 1]$.*

We present a lemma that bounds the difference between two integrals involving Gaussian kernels and a function $F(y)$, where the integrals are taken over different domains.

**Lemma 5.21** (Lemma 15 in (Fukumizu et al., 2025)). *Let $\alpha_t$ and $\beta_t$ be defined in Definition 3.10. Let $x_1 \in \mathbb{R}^d$ and $\epsilon \in (0, 0.1)$. Let $\omega := \exp(-\frac{\|x_1 - \beta_t y\|_2^2}{2\alpha_t^2})$. For any function $F(y)$ supported on $I^d$, there is $C_b > 0$ that depends only on $d$ such that*

$$|\int_{I^d} \frac{1}{(\sqrt{2\pi}\alpha_t)^d} \cdot \omega \cdot F(y)\mathrm{d}y - \int_{A_{x_1}} \frac{1}{(\sqrt{2\pi}\alpha_t)^d} \cdot \omega \cdot F(y)\mathrm{d}y| \le \epsilon,$$

where $A_{x_1} := \{y \in I^d \mid \|\frac{y-x_1}{\beta_t}\|_\infty \le C_b \frac{\alpha_t \sqrt{\log N}}{\beta_t}\}$.

## 5.6 FUNCTION COMPOSITION

We define the cardinal B-spline, a fundamental concept in approximation theory.

**Definition 5.22** (Implicited on Page 20 in (Fukumizu et al., 2025))**.** *Let $\mathcal{N}(x)$ be the function defined by $\mathcal{N}(x) = 1$ for $x \in [0,1]$ and 0 otherwise. The cardinal B-spline of order $\ell \in \mathbb{N}$ is defined by $\mathcal{N}_\ell(x) := \mathcal{N} * \mathcal{N} * \cdots * \mathcal{N}(x)$, which is the convolution $\ell$ times of $\mathcal{N}$.*

Building upon the definition of the cardinal B-spline, we now define the tensor product B-spline basis in $\mathbb{R}^d$.

**Definition 5.23** (Implicited on Page 21 in (Fukumizu et al., 2025))**.** *Let $\mathcal{N}_\ell(x)$ be defined in Definition 5.22 Then, we define the tensor product in B-spline basis in $\mathbb{R}^d$ of order $\ell$ as follows: $M_{k,j}^d(x) := \prod_{i=1}^d \mathcal{N}_{\ell+1}(2^{k_i} x_i - j_i)$.*

We present a theorem regarding the approximation of functions in Besov spaces using a linear combination of tensor product B-splines.

**Theorem 5.24** (Theorem 12 in (Fukumizu et al., 2025))**.** *Let $C > 0$ and $p', q', r \in \mathbb{R}_+$. Let $s > d \max\{1/p' - 1/r, 0\}$ and $0 < s < \min\{\ell, \ell - 1 + 1/p'\}$, where $\ell \in \mathbb{N}$ is the order of the cardinal B-spline bases. For any $f \in B_{p',q'}^s([-C,C]^d)$, there exists $f_N$ that satisfies*

$$\|f - f_N\|_{L_r([-C,C]^d)} \lesssim C^s N^{-s/d} \|f\|_{B_{p',q'}^s([-C,C]^d)}$$

*for $N \gg 1$, where $F_N(x)$ is defined as follows:*

$$f_N(x) = \sum_{k=0}^{K} \sum_{j \in J(k)} \mathcal{A}_{k,j} \cdot M_{k,j}^d(x) + \sum_{k=K+1}^{K^*} \sum_{i=1}^{n_k} \mathcal{A}_{k,j_i} \cdot M_{k,j_i}^d(x)$$

*with*

$$\sum_{k=0}^{K} |J(k)| + \sum_{k=K+1}^{K^*} n_k = N,$$

*where $J(k) = \{-C2^k - \ell, -C2^k - \ell + 1, \ldots, C2^k - 1, C2^k\}$, $(j_i)_{i=1}^{n_k} \subset J(k)$, $K = O(d^{-1} \log(N/C^d))$, $K^* = (O(1) + \log(N/C^d))\nu^{-1} + K$, $n_k = O((N/C^d)2^{-\nu(k-K)})$ $(k = K+1, \ldots, K^*)$ for $\nu = (s - \omega)/(2\omega)$ with $\omega = d \cdot \max\{1/p' - 1/r, 0\}$. Moreover, we can take $\mathcal{A}_{k,j}$ so that $|\mathcal{A}_{k,j}| \le N^{(\nu^{-1} + d^{-1}) \cdot \max\{d/p' - s, 0\}}$.*

## 6 CONCLUSION

In conclusion, we have provided a comprehensive theoretical framework for analyzing higher order matching in generative modeling and have rigorously established its worst case optimality as a distribution estimator. By leveraging refined techniques from flow matching generative frameworks, our analysis derives explicit upper bounds on the acceleration error of second-order flow matching across both small-$t$ and large-$t$ regimes. These bounds, expressed in terms of neural network complexity parameters such as depth, width, sparsity, and norm constraints and the intrinsic smoothness of the target density, which is quantified via Besov spaces, not only bridge the gap between first-order methods and their higher order counterparts but also highlight the statistical efficiency inherent in incorporating higher order corrections. Our results demonstrate that high-order flow matching can achieve nearly worst case optimal convergence rates under mild regularity assumptions, thereby offering a robust theoretical foundation for the design of fast, reliable, and statistically optimal generative algorithms. Moreover, the unified approach developed in this paper lays the groundwork for further investigations into advanced numerical techniques and neural network architectures that can further accelerate the sampling process while maintaining rigorous error control.

ETHIC STATEMENT

This paper does not involve human subjects, personally identifiable data, or sensitive applications. We do not foresee direct ethical risks. We follow the ICLR Code of Ethics and affirm that all aspects of this research comply with the principles of fairness, transparency, and integrity.

REPRODUCIBILITY STATEMENT

We ensure reproducibility of our theoretical results by including all formal assumptions, definitions, and complete proofs in the appendix. The main text states each theorem clearly and refers to the detailed proofs. No external data or software is required.

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

# Appendix

**Roadmap.** In Section A, we provide more related work. In Section B, we provide additional preliminaries. In Section C, we present full proofs of the theorem and lemmas in Section 5. In Section D, we present the detail proof for small $t$. In Section E, we present the detail proof for large $t$.

## A  MORE RELATED WORK

**Large Language Models.** The Transformer architecture (Vaswani et al., 2017) has swiftly risen to prominence as the leading framework for modern deep learning. The systems which scaled to billions of parameters and trained on extensive, diverse datasets are often labeled as large language models (LLMs) or foundation models (Bommasani et al., 2021; Chen et al., 2025d). Well-known examples of LLMs include Llama (Meta, 2024; Touvron et al., 2023), GPT4o (OpenAI, 2024), PaLM (Chowdhery et al., 2022), and BERT (Devlin et al., 2019), showcasing generalization capabilities (Bubeck et al., 2023) across a variety of downstream tasks. To adapt LLMs for specialized domains, researchers have developed a range of techniques. These encompass: adapter modules (Shi et al., 2023a; Zhang et al., 2023b; Gao et al., 2023a; Hu et al., 2022); calibration methods (Zhao et al., 2021; Zhou et al., 2023); multitask fine-tuning (Xu et al., 2023; 2024d; Von Oswald et al., 2023; Gao et al., 2021); as well as prompt engineering (Lester et al., 2021), scratchpad strategies (Nye et al., 2021), instruction tuning (Mishra et al., 2022; Li & Liang, 2021; Chung et al., 2022), symbolic adaptation (Xu et al., 2022b; 2024b; Wei et al., 2023), black-box tuning (Sun et al., 2022), reinforcement learning aligned with human feedback (Ouyang et al., 2022), and structured reasoning methods (Zheng et al., 2024; Wei et al., 2022; Yao et al., 2023; Khattab et al., 2022). Recent studies also explore advancements in tensor architectures (Zhang et al., 2025; Sanford et al., 2024; Liang et al., 2024e; Alman & Song, 2024a; Ke et al., 2024b), efficiency improvements (Xu et al., 2024a; Ke et al., 2024a; Hu et al., 2023; 2024f;c;b; Wu et al., 2024b;a; Song et al., 2024; Shen et al., 2024; 2025a; Hu et al., 2024a; Chen et al., 2024b;c; Alman & Song, 2024b; Shi et al., 2024a; Liang et al., 2024c; Li et al., 2024b;e;d;c; Chen et al., 2025b; Liang et al., 2024a; Li et al., 2024c), and other supplementary research (Li et al., 2025b; Ke et al., 2025; Chen et al., 2025f; Zhang et al., 2024a; Zhang, 2024; Xu et al., 2024c; Xie et al., 2022; Tan et al., 2023; Song & Yang, 2023; Sinha et al., 2023; Li et al., 2025a; Liang et al., 2025a; Wu et al., 2024c; Hu et al., 2024d;e; Gao et al., 2023c; Chen et al., 2025e;b; Liang et al., 2025b; Deng et al., 2022; Chang et al., 2024; Demirel et al., 2022; Chen et al., 2024a; Shrivastava et al., 2023; Li et al., 2024f; Liang et al., 2024d; Li et al., 2025c; 2024a; Gao et al., 2023d;b; Liang et al., 2025b; 2024b; Chen et al., 2025e; Wang et al., 2024a; Shi et al., 2024b; 2023b;c).

## B  ADDITIONAL PRELIMINARY

### B.1  BESOV SPACE

To quantify the smoothness of functions, we utilize the $r$-th modulus of smoothness, defined as follows:

**Definition B.1** ($r$-th Modulus of Smoothness, Definition 2.2 on Page 3 in (Oko et al., 2023)). *If the following conditions hold:*

- $p \in (0, \infty]$.

- $f \in L^p(\Omega)$.

*We define the $r$-th modulus of smoothness of $f$ as*

$$w_{r,p}(f, t) := \sup_{\|h\|_2 \leq t} \|\Delta_h^r(f)\|_p,$$

*where*

$$\Delta_h^r(f)(x) := \begin{cases} \sum_{j=0}^r \binom{r}{j} \cdot (-1)^{r-j} \cdot f(x + jh) & \text{if } x + jh \in \Omega \text{ for all } j; \\ 0 & \text{otherwise.} \end{cases}$$

Building upon the concept of the modulus of smoothness, we now introduce the Besov space $B_{p,q}^s(\Omega)$, which provides a more refined classification of function smoothness.

**Definition B.2** (Besov space $B_{p,q}^s(\Omega)$, Definition 2.3 on page 3 in (Oko et al., 2023))**.** *If the following conditions hold:*

- *$p > 0$.*

- *$q \leq \infty$.*

- *$s > 0$.*

- *Let $r$ be defined as $r := \lfloor s \rfloor + 1$.*

- *Let the $r$-th modulus of smoothness of $f$, i.e., $w_{r,p}(f,t)$, be defined in Definition B.1.*

*We define the Besov space $B_{p,q}^s$ as*
$$B_{p,q}^s := \{f \in L^p(\Omega) \mid \|f\|_{B_{p,q}^s} < \infty\}.$$
*We define the seminorm of Besov space $B_{p,q}^s$ as*
$$|f|_{B_{p,q}^s} = \begin{cases} (\int_0^\infty (t^{-s} w_{r,p}(f,t))^q \frac{\mathrm{d}t}{t})^{\frac{1}{q}} & \text{if } q < \infty; \\ \sup_{t>0}\{t^{-s} w_{r,p}(f,t)\} & \text{if } q = \infty. \end{cases}$$
*We define the norm of Besov space $B_{p,q}^s$ as*
$$\|f\|_{B_{p,q}^s} := \|f\|_p + |f|_{B_{p,q}^s}$$

## B.2 FIRST ORDER ERROR BOUND

Here, we present the preliminary result here to show the error bound of first order flow matching.

**Lemma B.3** (Theorem 7 in (Fukumizu et al., 2025))**.** *If the following conditions hold:*

- *Assume Assumption 5.2, 5.3, 5.4, 5.5, 5.7, 5.9 hold.*

- *Let $C_6$ be a constant independent of $t$.*

- *Let $\alpha_t$ and $\beta_t$ be defined in Definition 3.10.*

*Then there is a neural network $\phi_1 \in \mathcal{M}(L, W, S, B)$, such that, for sufficiently large $N$,*
$$\int \|\phi_1(x_1, t) - v_t(x_1)\|_2^2 \cdot p_t(x_1)\mathrm{d}x_1 \leq C_6 \cdot (\alpha_t'^2 \log N + \beta_t'^2) \cdot N^{-\frac{2s}{d}},$$
*for any $t \in [T_0, 3T_*]$, where*
$$L = O(\log^4 N), \|W\|_\infty = O(N \log^6 N), S = O(N \log^8 N), B = \exp(O(\log N \log\log N)).$$

**Lemma B.4** (Theorem 8 in (Fukumizu et al., 2025))**.** *If the following conditions hold:*

- *Fix $t_* \in [T_*, 1]$ and take arbitrary $\eta > 0$.*

- *Assume Assumption 5.2, 5.3, 5.4, 5.5, 5.7, 5.9 hold.*

- *Let $C_7 > 0$ be a constant independent of $t$.*

- *Let $t_* \in [T_*, 1]$.*

- *Let $\alpha_t$ and $\beta_t$ be defined in Definition 3.10.*

*Then there is a neural network $\phi_2 \in \mathcal{M}(L, W, S, B)$, such that the bound*
$$\int \|\phi_2(x_1, t) - v_t(x_1)\|_2^2 \cdot p_t(x_1)\mathrm{d}x_1 \leq C_7 \cdot (\alpha_t'^2 \log N + \beta_t'^2) \cdot N^{-\eta}$$
*holds for any $t \in [2t_*, 1]$, where*
$$L = O(\log^4 N), \|W\|_\infty = O(N), S = O(t_*^{-d\kappa} N^{\delta\kappa}), B = \exp(O(\log N \log\log N))$$

## C  PROOF OF THEOREM AND TECHNICAL LEMMAS

In this section, we provide detailed proofs of the theorems and technical lemmas presented in Section 5.

**Theorem C.1** (Bound of $a_t$, formal version of Theorem 5.11). *If the following conditions hold:*

- *Let $x_1$, $\alpha$, $\beta$ be defined in Definition 3.10.*

- *Let $C_3 > 0$ be a constant depend on $d$ and $C_0$.*

*Then we can show that*

$$\|a_t(x_1)\|_2 \leq C_3 \cdot (\alpha_t'' \cdot \max\{(\|x_1\|_\infty - \beta_t)/\alpha_t, 1\} + |\beta_t''|)$$

*for any $x_1 \in \mathbb{R}^d$ and $t \in [T_0, 1]$.*

*Proof.* Differentiating $v_t(x_1 \mid y) = \alpha_t' \frac{x_1 - \beta_t y}{\alpha_t} + \beta_t' y$, we obtain

$$a_t(x_1 \mid y) = \frac{dv_t(x_1 \mid y)}{dt}$$

$$= \alpha_t'' \cdot \frac{x_1 - \beta_t y}{\alpha_t} + \alpha_t' \cdot \left(\frac{-\beta_t' y \alpha_t - (x_1 - \beta_t y)\alpha_t'}{\alpha_t^2}\right) + \beta_t'' y$$

$$= \alpha_t'' \cdot \frac{x_1 - \beta_t y}{\alpha_t} + \beta_t'' y - \alpha_t'^2 \cdot \frac{x_1 - \beta_t y}{\alpha_t^2} - \alpha_t' \cdot \frac{\beta_t' y}{\alpha_t}.$$

This leads to the upper bound

$$\|a_t(x_1)\|_2$$

$$\leq \left(\int \|\alpha_t'' \cdot \frac{x_1 - \beta_t y}{\alpha_t} + \beta_t'' y - \alpha_t'^2 \cdot \frac{x_1 - \beta_t y}{\alpha_t^2} - \alpha_t' \cdot \frac{\beta_t' y}{\alpha_t}\|_2 \cdot g(x_1; \beta_t y, \alpha_t^2) p_0(y) dy\right)/p_t(x_1).$$

Splitting into separate terms, we get

$$\|\alpha_t'' \cdot \frac{x_1 - \beta_t y}{\alpha_t}\|_2 \leq C_{1,a} \cdot \alpha_t'' \cdot \max\{(\|x_1\|_\infty - \beta_t)/\alpha_t, 1\},$$

$$\|\alpha_t'^2 \cdot \frac{x_1 - \beta_t y}{\alpha_t^2}\|_2 \leq C_{2,a} \cdot \alpha_t' \cdot \max\{(\|x_1\|_\infty - \beta_t)/\alpha_t, 1\},$$

$$\|\beta_t'' y\|_2 \leq |\beta_t''|,$$

$$\|\alpha_t' \cdot \frac{\beta_t' y}{\alpha_t}\|_2 \leq |\frac{\beta_t' \cdot \alpha_t'}{\alpha_t}|$$

where $C_{1,a}$ and $C_{2,a}$ are some constants which are bounded by $C_3$.

Combining these bounds, we conclude

$$\|a_t(x_1)\|_2 \leq C_3 \cdot (\alpha_t'' \cdot \max\{(\|x_1\|_\infty - \beta_t)/\alpha_t, 1\} + |\beta_t''|).$$

This completes the proof. □

**Lemma C.2** (Bound $a_t$ with Constant, formal version of Lemma 5.12). *Let $\epsilon \in (0, 0.1)$ be a small number. Let $\alpha_t$ and $\beta_t$ be defined in Definition 3.10. For any $C_4 > 0$, we have*

$$\|a_t(x_1)\|_2 \leq C_4 \cdot (\alpha_t'' \sqrt{\log(1/\epsilon)} + |\beta_t''|)$$

*for any $x_1$ with $\|x_1\|_\infty \leq \beta_t + C_4 \alpha_t \sqrt{\log(1/\epsilon)}$ and $t \in [T_0, 1]$.*

*Proof.* The proof simply follows from $\|x_1\|_\infty \leq \beta_t + C_4 \alpha_t \sqrt{\log(1/\epsilon)}$ and Lemma 5.11. □

**Lemma C.3** (Omit Small Term of Integral on Acceleration, formal version of Lemma 5.13). *If the following conditions hold:*

- *Let $\omega > 0$ be an arbitrary positive number.*

- *Let $u$ be a neural network.*

- *Let $C' > 0$ be a constant.*

- *Let $D := \{x_1 \in \mathbb{R}^d \mid \|x_1\|_\infty \leq \beta_t + C_4 \alpha_t \sqrt{\log N}\}$*

- *Let $x_1$ be the trajectory, $x_2 := \phi_2(x_1, t)$ where $\phi_2$ is the neural network in Lemma B.4.*

- *Let $x$ be defined as the concatenation of $x_1$ and $x_2$, i.e., $x := [x_1, x_2]$.*

- *Let $\alpha_t$ and $\beta_t$ be defined in Definition 3.10.*

*Then we have*

$$\int_D \mathbf{1}[p_t(x_1) \leq N^{-\frac{2s+\omega}{d}}] \cdot \|a_t(x_1) - u(x,t)\|_2^2 \cdot p_t(x_1)\mathrm{d}x_1 = o(C' \cdot (\alpha_t''^2 \log N + \beta_t''^2)N^{-\frac{2s}{d}})$$

*Proof.* We want to proof this term as a small term such that in the proof of Theorem D.1, we can omit it.

$$\int_D \mathbf{1}[p_t(x_1) \leq N^{-\frac{2s+\omega}{d}}] \cdot \|a_t(x_1) - u(x,t)\|_2^2 \cdot p_t(x_1)\mathrm{d}x_1$$

$$\leq 4C_3^2 \int_D ((\alpha_t'')^2 \log N + |\beta_t''|^2)N^{-\frac{2s+\omega}{d}}\mathrm{d}x_1$$

$$\leq 4C_3^2 \cdot N^{-\frac{2s+\omega}{d}} \cdot ((\alpha_t'')^2 \log N + |\beta_t''|^2) \cdot 2^d \cdot (\beta_t + C_4 a_t \sqrt{\log N})^d$$

$$\leq C'' \cdot ((\alpha_t'')^2 \log N + |\beta_t''|^2) \cdot N^{-\frac{2s+\omega}{d}} \cdot \log^{\frac{d}{2}} N,$$

where $C''$ is a small constant that $C'' = o(C')$. Thus, we finish the proof. $\square$

**Lemma C.4** (Bound Gamma Function, formal version of Lemma 5.14)**.** *If the following conditions hold:*

- *Let $\ell \in \mathbb{N}$.*

- *Let $\psi_\ell(z) := \int_z^\infty r^\ell \exp(-\frac{r^2}{2})\mathrm{d}r$.*

*Then we have the bound that:*

$$\psi_\ell(z) \leq \ell!! \cdot z^{\ell-1} \cdot \exp(-\frac{z^2}{2})$$

*Proof.* To find an upper bound for $\psi_\ell(z) = \int_z^\infty r^\ell \exp(-\frac{r^2}{2})\mathrm{d}r$ for $z \geq 1$, we use integration by parts and induction.

First, we establish the recursive relationship for $\psi_\ell(z)$:

$$\psi_\ell(z) = z^{\ell-1} \cdot \exp(-\frac{z^2}{2}) + (\ell - 1) \cdot \psi_{\ell-2}(z)$$

Using this recursion, we can derive the bound by induction.

**Base Cases:**

- For $\ell = 1$, $\psi_1(z) = \exp(-\frac{z^2}{2})$, which is trivially bounded by $1!! \cdot z^0 \cdot \exp(-\frac{z^2}{2})$.

- For $\ell = 2$, $\psi_2(z) = z \cdot \exp(-\frac{z^2}{2}) + \psi_0(z)$. Since $\psi_0(z) \leq \exp(-\frac{z^2}{2})/z$ for $z \geq 1$, we get $\psi_2(z) \leq 2z \cdot \exp(-\frac{z^2}{2})$.

**Inductive Step:** Assume for all $k < \ell$, $\psi_k(z) \leq k!! \cdot z^{k-1} \cdot \exp(-\frac{z^2}{2})$. For $\ell$, we use the recursion:

$$\psi_\ell(z) = z^{\ell-1} \cdot \exp(-\frac{z^2}{2}) + (\ell - 1) \cdot \psi_{\ell-2}(z)$$

By the induction hypothesis, $\psi_{\ell-2}(z) \leq (\ell-2)!! \cdot z^{\ell-3} \exp(-\frac{z^2}{2})$. Substituting this into the recursion gives:

$$\psi_\ell(z) \leq z^{\ell-1} \cdot \exp(-\frac{z^2}{2}) + (\ell-1)(\ell-2)!! \cdot z^{\ell-3} \cdot \exp(-\frac{z^2}{2})$$

$$= \exp(-\frac{z^2}{2}) \cdot z^{\ell-3} \cdot (z^2 + (\ell-1)(\ell-2)!!)$$

Since $(\ell-1)(\ell-2)!! \leq \ell!!$, we get:

$$\psi_\ell(z) \leq \ell!! \cdot z^{\ell-1} \cdot \exp(-\frac{z^2}{2})$$

$\square$

**Lemma C.5** (Bounded Integral $a_t$, formal version of Lemma 5.15). *Let $\alpha_t$ and $\beta_t$ be defined in Definition 3.10. For any $C_5 > 0$, there is $\widetilde{C} > 0$ such that*

$$\int_{\|x_1\|_\infty \geq \beta_t + C_5 \alpha_t \sqrt{\log(1/\epsilon)}} p_t(x_1) \cdot \|a_t(x_1)\|_2^2 dx_1 \leq \widetilde{C} \cdot \epsilon^{\frac{C_5^2}{2}} \cdot (\alpha_t''^2 \log^{\frac{d}{2}}(1/\epsilon) + \beta_t''^2 \log^{\frac{d-2}{2}}(1/\epsilon))$$

*hold for any $\epsilon > 0$ and $t \in [T_0, 1]$.*

*Proof.* Follows from Lemmas 5.20 and 5.11, we have

$$p_t(x_1) \cdot \|a_t(x_1)\|_2^2$$

$$\leq 2C_1 C_3 \cdot \exp(-0.5 \max\{\|x_1\|_\infty - \beta_t, 0\}^2 / \alpha_t^2) \cdot (\alpha_t''^2 \max\{\|x_1\|_\infty - \beta_t, 0\}^2 / \alpha_t^2 + \beta_t''^2).$$

where $C_1$ is a constant in Lemma 5.20, $C_3$ is a constant in Theorem 5.11.

Let $r := \max\{\|x_1\|_\infty - \beta_t, 0\}/\alpha_t$. Let $B_i := \{x_1 = (x_{1,1}, \ldots, x_{1,d}) \in \mathbb{R}^d \mid |x_{1,i}| = \max_{1 \leq j \leq d} |x_{1,j}|\}$. In $B_1$, the variables $x_{1,2}, \ldots, x_{1,d}$ satisfy $|x_{1,j}| \leq \beta_t + C_4 \alpha_t \sqrt{\log(1/\epsilon)}$. Note that

$$(\alpha_t r + \beta_t)^{d-1} \leq (r+1)^{d-1} \leq 2^{d-1} r^{d-1}$$

where $C_4$ is a constant in Lemma 5.12, the first step follows from that $\alpha_t, \beta_t \in [0, 1]$, and the second step follows from that $\sum_{i=1}^{d-1} \binom{d-1}{i} \leq 2^{d-1}$. Thus we have

$$2C_1 C_3 d \cdot \int_{C_4 \sqrt{\log(1/\epsilon)}} \exp(-\frac{r^2}{2}) \cdot (\alpha_t''^2 r^2 + \beta_t''^2)(\alpha_t r + \beta_t)^{d-1} dr$$

$$\leq C' \int_{C_4 \sqrt{\log(1/\epsilon)}} \exp(-\frac{r^2}{2}) \cdot (\alpha_t''^2 r^{d+1} + \beta_t''^2 r^{d-1}) dr,$$

which follows from $(\alpha_t r + \beta_t)^{d-1} \leq (r+1)^{d-1} \leq 2^{d-1} r^{d-1}$, where $C_1$ is a constant in Lemma 5.20, $C_3$ is a constant in Theorem 5.11, $C_4$ is a constant in Lemma 5.12.

For $\ell \in \mathbb{N}$, we define $\psi_\ell(z) := \int_z^\infty r^\ell \exp(-\frac{r^2}{2}) dr$. Following from Lemma 5.14, we can show

$$\psi_\ell(z) \leq B_\ell z^{\ell-1} \cdot \exp(-\frac{z^2}{2}),$$

where $B_\ell$ is a constant only depend on $\ell$.

Thus we obtain an upper bound

$$\int_{\|x_1\|_\infty \geq \beta_t + C_5 \alpha_t \sqrt{\log(1/\epsilon)}} p_t(x_1) \cdot \|a_t(x_1)\|_2^2 dx_1 \leq \widetilde{C} \cdot \epsilon^{\frac{C_5^2}{2}} \cdot (\alpha_t''^2 \log^{\frac{d}{2}}(1/\epsilon) + \beta_t''^2 \log^{\frac{d-2}{2}}(1/\epsilon)),$$

it follows from Lemma 5.12. Thus, we finish the proof. $\square$

## D  BOUNDS ON SECOND ORDER FLOW MATCHING SMALL $t$

We formally present the proof of Theorem 4.1 in this section.

**Theorem D.1** (Main Theorem, Bound Acceleration Error under Small $t$, Formal Version of Theorem 4.1). *If the following conditions hold:*

- *Assume Assumption 5.2, 5.3, 5.4, 5.6, 5.8, 5.9 hold.*

- *Let $C_6$ be a constant independent of $t$.*

- *Let $x_1$ be the trajectory, $x_2 := \phi_1(x_1, t)$ where $\phi_1$ is the neural network in Lemma B.3.*

- *Let $x$ be defined as the concatenation of $x_1$ and $x_2$, i.e., $x := [x_1, x_2]$.*

*then there is a neural network $u_1 \in \mathcal{M}(L, W, S, B)$ and a constant $C$, which is independent of $t$, such that, for sufficiently large $N$,*

$$\int \|u_1(x, t) - a_t(x_1)\|_2^2 \cdot p_t(x_1)\mathrm{d}x_1 \leq C_6 \cdot (\alpha_t''^2 \log N + \beta_t''^2) \cdot N^{-\frac{2s}{d}},$$

*for any $t \in [T_0, 3T_*]$, where*

$$L = O(\log^4 N), \|W\|_\infty = O(N \log^6 N), S = O(N \log^8 N), B = \exp(O((\log N) \cdot (\log \log N))).$$

*Proof.* First, we show that the LHS can be approximated by the integral on the bounded region:

$$D := \{x \in \mathbb{R}^d \mid \|x\|_\infty \leq \beta_t + C_4 \alpha_t \sqrt{\log N}\}$$

where $C_4$ is a constant in Lemma 5.12.

We can bound $a_t(x_1)$ when $x \in D$:

$$\|a_t(x_1)\|_2 \leq C_4 \cdot (\alpha_t''^2 \log N + \beta_t''^2)$$

which follows from Lemma 5.11.

Since $\|a_t(x_1)\|_2^2$ is bounded, we think $\|u_1(x, t)\|_2^2$ is also bounded by the same term.

$$\int_D \|u_1(x, t) - a_t(x_1)\|_2^2 \cdot p_t(x_1)\mathrm{d}x_1$$

$$\leq 2C_4 \cdot (\alpha_t''^2 \log N + \beta_t''^2) \cdot \int_D p_t(x_1)\mathrm{d}x_1$$

$$\leq 2C_4 \cdot (\alpha_t''^2 \log N + \beta_t''^2) \cdot \int_D \frac{1}{(\sqrt{2\pi\alpha_t})^d} \cdot \exp(-\frac{\|x_1 - \beta_t y\|_2^2}{2\alpha_t^2})\mathrm{d}x_1$$

$$\leq 2C_4 \cdot (\alpha_t''^2 \log N + \beta_t''^2) \cdot \widetilde{C} \cdot N^{-C_4^2/2} \cdot \log^{\frac{d-2}{2}} N.$$

where the first step follows from Lemma 5.11, the second step follows from the distribution of $p_t(x_1)$, the third step follows from Lemma 5.15.

Further, we can show that:

$$\int \|u_1(x, t) - a_t(x_1)\|_2^2 \cdot p_t(x_1)\mathrm{d}x_1$$

$$\leq \int_D \|u_1(x, t) - a_t(x_1)\|_2^2 \cdot p_t(x_1)\mathrm{d}x_1 + C' \cdot (\alpha_t''^2 \log N + \beta_t''^2) \cdot N^{-\frac{2s}{d}}. \tag{1}$$

which follows from $C_4$ can be selected large enough, i.e., $C_4^2/2 > \frac{2s}{d}$. Here $C' > 0$ is some constant.

For the first term of Eq. (1), we have:

$$\int_D \|u_1(x, t) - a_t(x_1)\|_2^2 \cdot p_t(x_1)\mathrm{d}x_1$$

$$= \int_D \mathbf{1}[p_t(x_1) \geq N^{-\frac{2s+\omega}{d}}] \cdot \|u_1(x,t) - a_t(x_1)\|_2^2 \cdot p_t(x_1)\mathrm{d}x_1$$

$$+ \int_D \mathbf{1}[p_t(x_1) \leq N^{-\frac{2s+\omega}{d}}] \cdot \|u_1(x,t) - a_t(x_1)\|_2^2 \cdot p_t(x_1)\mathrm{d}x_1$$

Following from Lemma 5.13, we can omit the second term. Combine with Eq. (1), we have:

$$\int \|u_1(x,t) - a_t(x_1)\|_2^2 \cdot p_t(x_1)\mathrm{d}x_1$$

$$\leq \int_D \mathbf{1}[p_t(x_1) \geq N^{-\frac{2s+\omega}{d}}] \cdot \|u_1(x,t) - a_t(x_1)\|_2^2 \cdot p_t(x_1)\mathrm{d}x_1 + C' \cdot (\alpha_t''^2 \log N + \beta_t''^2) \cdot N^{-\frac{2s}{d}}.$$

Following the property of flow matching, we have the following:

$$a_t(x_1) = \frac{\int a_t(x_1 \mid y) \cdot p_t(x_1 \mid y) \cdot p_0(y)\mathrm{d}y}{p_t(x_1)}, \ p_t(x_1) = \int p_t(x_1 \mid \widetilde{y}) \cdot p_0(y)\mathrm{d}y. \tag{2}$$

Base on Lemma 5.16, there is a $F_N$ such that:

$$\|p_0 - f_N\|_{L^2(I^d)} \leq C_a N^{-s/d}, \ \|p_0 - f_N\|_{L^2(I^d \setminus I_N^d)} \leq C_a N^{-\widetilde{s}/d}$$

where $C_a$ is a constant in Lemma 5.16.

We define an approximate of $p_t(x_1)$, which is $\widetilde{f}(x_1, t)$, as follows:

$$\widetilde{f}(x_1, t) := \int \frac{1}{(\sqrt{2\pi}\alpha_t)^d} \cdot \exp(-\frac{\|x_1 - \beta_t y\|_2^2}{2\alpha_t}) \cdot f_N(y)\mathrm{d}y$$

Now we define 3 different functions. We define $f_1$ as follows:

$$f_1(x_1, t) := \max\{\widetilde{f}(x_1, t), N^{-\frac{2s+\omega}{d}}\}.$$

We define $f_2$ and $f_3$ in order to construct a function to approximate the numerator of Eq. (2) as follows:

$$f_2(x_1, t) := \int \frac{x_1 - \beta_t y}{\alpha_t} \cdot \frac{1}{(\sqrt{2\pi}\alpha_t)^d} \cdot \exp(-\frac{\|x_1 - \beta_t y\|_2^2}{2\alpha_t^2}) \cdot f_N(y)\mathrm{d}y,$$

$$f_3(x_1, t) := \int y \frac{1}{(\sqrt{2\pi}\alpha_t)^d} \cdot \exp(-\frac{\|x_1 - \beta_t y\|_2^2}{2\alpha_t^2}) \cdot f_N(y)\mathrm{d}y.$$

Further, we construct a function to approximate the numerator of Eq. (2) base on $f_2(x_1, t)$ and $f_3(x_1, t)$:

$$\alpha_t'' f_2(x_1, t) + \beta_t'' f_3(x_1, t).$$

Base on $f_1$, $f_2$ and $f_3$, we construct $f_4$ to approximate $v_t(x_1)$:

$$f_4(x_1, t) := \frac{\alpha_t'' f_2(x_1, t) + \beta_t'' f_3(x_1, t)}{f_1(x_1, t)} \cdot \mathbf{1}[|\frac{f_2(x_1, t)}{f_1(x_1, t)}| \leq C_5 \sqrt{\log N}] \cdot \mathbf{1}[|\frac{f_3(x_1, t)}{f_1(x_1, t)}| \leq C_5],$$

where $C_5$ is a constant in 5.15.

We can split the integral

$$\int_D \mathbf{1}[p_t(x_1) \geq N^{-\frac{2s+\omega}{d}}] \cdot \|u_1(x,t) - a_t(x_1)\|_2^2 \cdot p_t(x_1)\mathrm{d}x_1$$

$$\leq \int_D \mathbf{1}[p_t(x_1) \geq N^{-\frac{2s+\omega}{d}}] \cdot \|u_1(x,t) - f_4(x_1,t)\|_2^2 \cdot p_t(x_1)\mathrm{d}x_1$$

$$+ \int_D \mathbf{1}[p_t(x_1) \geq N^{-\frac{2s+\omega}{d}}] \cdot \|f_4(x_1,t) - a_t(x_1)\|_2^2 \cdot p_t(x_1)\mathrm{d}x_1$$

$$=: I_A + I_B. \tag{3}$$

which follows from triangle inequality. In the following part, we bound $I_A$ and $I_B$ separately.

**Bound of $I_A$.**

Using neural networks, we approximate $f_1$, $f_2$, and $f_3$. Here, we adopt a minor notational abuse by interpreting $u$ as a scalar whenever it appears without parentheses, rather than as a neural network. For $k \in \mathbb{Z}_+$ and $j \in \mathbb{Z}^d$, let $E_{k,j,u,a}$ ($a = 1, 2, 3$, $u = 0, 1$) denote the function defined by:

$$E_{k,j,u,1} := \int_{\mathbb{R}^d} \mathbf{1}[\|y\|_\infty \le C_{b,1}] \cdot M_{k,j}^d(y) \cdot \frac{1}{(\sqrt{2\pi}\alpha_t)^d} \cdot \exp(-\frac{\|x_1 - \beta_t y\|_2^2}{2\alpha_t^2}) \mathrm{d}y,$$

$$E_{k,j,u,2} := \int_{\mathbb{R}^d} \frac{x_1 - \beta_t y}{\alpha_t} \cdot \mathbf{1}[\|y\|_\infty \le C_{b,1}] \cdot M_{k,j}^d(y) \cdot \frac{1}{(\sqrt{2\pi}\alpha_t)^d} \cdot \exp(-\frac{\|x_1 - \beta_t y\|_2^2}{2\alpha_t^2}) \mathrm{d}y,$$

$$E_{k,j,u,3} := \int_{\mathbb{R}^d} y \cdot \mathbf{1}[\|y\|_\infty \le C_{b,1}] \cdot M_{k,j}^d(y) \cdot \frac{1}{(\sqrt{2\pi}\alpha_t)^d} \cdot \exp(-\frac{\|x_1 - \beta_t y\|_2^2}{2\alpha_t^2}) \mathrm{d}y.$$

where $C_{b,1}$ is defined as:

$$C_{b,1} := \begin{cases} 1 & \text{if } u = 0; \\ 1 - N^{-\frac{\kappa^{-1}-\delta}{d}} & \text{if } u = 1; \end{cases}$$

Base on Theorem 5.24, $f_N$ is written as a linear combination of $\mathbf{1}[\|y\|_2 \le C_{b,1}]M_{k,j}^d$ with coefficients $\mathcal{A}_{k,j}$. Here, $M_{k,j}^d$ is given by Definition 5.23.

Following from Lemma 5.16, for any $\epsilon > 0$, there are neural network $u_5, u_6, u_7$ such that:

$$|f_1(x_1, t) - u_5(x, t)| \le D_5 N \max_{i \in [N]} |\mathcal{A}_i|\epsilon$$

$$\|f_2(x_1, t) - u_6(x, t)\|_2 \le D_6 N \max_{i \in [N]} |\mathcal{A}_i|\epsilon,$$

$$\|f_3(x_1, t) - u_7(x, t)\|_2 \le D_7 N \max_{i \in [N]} |\mathcal{A}_i|\epsilon.$$

Since $\max_i |\mathcal{A}_i| \le N^{-(\sqsubseteq^{-1}+d^{-1})(d/p-s)}$, by taking $\epsilon$ sufficently small, for any $\eta > 0$, we have:

$$|f_1(x_1, t) - u_5(x, t)| \le D_5 N^{-\eta}$$

$$\|f_2(x_1, t) - u_6(x, t)\|_2 \le D_6 N^{-\eta}$$

$$\|f_3(x_1, t) - u_7(x, t)\|_2 \le D_7 N^{-\eta}.$$

Then, we perform the following operation to approximate $v_t(x_1)$:

$$\zeta_1 := \mathsf{clip}(u_5; N^{-(2s+\omega)/d}, N^{K_0+1}),$$
$$\zeta_2 := \mathsf{recip}(\zeta_1),$$
$$\zeta_3 := \mathsf{mult}(\zeta_2, u_6),$$
$$\zeta_4 := \mathsf{clip}(\zeta_3; -C_5\sqrt{\log N}, C_5\sqrt{\log N}),$$
$$\zeta_5 := \mathsf{mult}(\zeta_2, u_7),$$
$$\zeta_6 := \mathsf{clip}(\zeta_5; -C_5, C_5),$$
$$\zeta_7 := \mathsf{mult}(\zeta_4, \widehat{a}'),$$
$$\zeta_8 := \mathsf{mult}(\zeta_6, \widehat{b}'),$$
$$u_8 := \zeta_7 + \zeta_8.$$

where clip is the neural network in Lemma 5.17, recip is the neural network in Lemma 5.18, and mult is the neural network in Lemma 5.19.

Following from Lemma 5.17, 5.18, and 5.19, the neural networks clip, recip, and mult can simulate the clip operation, the reciprocal function, and the operation of product of $d$ scalars with the error of $N^{-\eta}$ for arbitrarily large $\eta$. Also, their overall complexity is bounded by

$$\mathrm{poly}(N + B + \|W\|_\infty),$$

while the height $L$ and sparsity constraint $S$ are bounded by

$$\text{poly}(\log N).$$

Considering the upper bound error, the parameters $B$, $\|W\|_\infty$ are bounded by the $\log(C/\epsilon)$ term with some constant $C$.

Furthermore, we want to approximate $\alpha_t'$ and $\beta_t'$ by $\widehat{\alpha}_t'$ and $\widehat{\beta}_t'$ using neural networks in the construction. Following the similar method with Section B of (Oko et al., 2023), the network parameters can be bounded by

$$O(\log^r(1/\epsilon))$$

with approximation accuracy of $\epsilon$.

Consequently, extending the neural networks to derive $\phi_8$ from $\phi_5$, $\phi_6$, and $\phi_7$ increases the log covering number merely by a factor of $\text{poly}(\log N)$.

**Bound of $I_B$.**

By definition of $I_B$, we have:

$$I_B = \int_D \mathbf{1}[p_t(x_1) \geq N^{-\frac{2s+\omega}{d}}] \cdot \|f_4(x_1, t) - a_t(x_1)\|_2^2 \cdot p_t(x_1) \mathrm{d}x_1$$

We define $h_2$ and $h_3$ as follows:

$$h_2(x_1, t) := \int_{\mathbb{R}^d} \frac{x_1 - \beta_t y}{\alpha_t} \cdot \frac{1}{(\sqrt{2\pi}\alpha_t)^d} \cdot \exp(-\frac{\|x_1 - \beta_t y\|_2^2}{2\alpha_t^2}) \cdot p_0(y) \mathrm{d}y,$$

$$h_3(x_1, t) := \int_{\mathbb{R}^d} y \cdot \frac{1}{(\sqrt{2\pi}\alpha_t)^d} \cdot \exp(-\frac{\|x_1 - \beta_t y\|_2^2}{2\alpha_t^2}) \cdot p_0(y) \mathrm{d}y.$$

base on $h_2$ and $h_3$, we have

$$\|f_4(x_1, t) - a_t(x_1)\|_2$$
$$= \mathbf{1}[\|\frac{f_2(x_1, t)}{f_1(x_1, t)}\|_2 \leq C_5\sqrt{\log N}] \cdot \mathbf{1}[\|\frac{f_3(x_1, t)}{f_1(x_1, t)}\|_2 \leq C_5]$$
$$\cdot \|\frac{\alpha_t'' f_2(x_1, t) + \beta_t'' f_3(x_1, t)}{f_1(x_1, t)} - \frac{\alpha_t'' h_2(x_1, t) + \beta_t' h_3(x_1, t)}{p_t(x_1)}\|_2.$$

In the following part, we divide the region $D$ into 2 parts:

- **Part 1.** $\|x_1\|_\infty \leq \beta_t$

- **Part 2.** $\beta_t \leq \|x_1\|_\infty \leq \beta_t + C_4$

Recall that we use $f_1(x_1, t)$ to approximate $p_t(x_1)$, and we have $p_t(x_1) \geq N^{-(2s+\omega)/d}$, we also assume that $f_1(x_1, t) \geq N^{-(2s+\omega)/d}$.

**Proof of Part 1.** $\|x_1\|_\infty \leq \beta_t$

Following from Lemma 5.20, we know $C_1^{-1} \leq p_t(x_1) \leq C_1$. Thus we have

$$\|\frac{\alpha_t'' f_2(x_1, t) + \beta_t'' f_3(x_1, t)}{f_1(x_1, t)} - \frac{\alpha_t'' h_2(x_1, t) + \beta_t'' h_3(x_1, t)}{p_t(x_1)}\|_2$$
$$\leq |\alpha_t''| \cdot \|\frac{f_2(x_1, t)}{f_1(x_1, t)} - \frac{h_2(x_1, t)}{p_t(x_1)}\|_2 + |\beta_t''| \cdot \|\frac{f_3(x_1, t)}{f_1(x_1, t)} - \frac{h_3(x_1, t)}{p_t(x_1)}\|_2$$
$$\leq |\alpha_t''| \cdot (\|\frac{f_2(x_1, t)}{f_1(x_1, t)} - \frac{f_2(x_1, t)}{p_t(x_1)}\|_2 + \|\frac{f_2(x_1, t)}{p_t(x_1)} - \frac{h_2(x_1, t)}{p_t(x_1)}\|_2)$$
$$+ |\beta_t''| \cdot (\|\frac{f_2(x_1, t)}{f_1(x_1, t)} - \frac{f_3(x_1, t)}{p_t(x_1)}\|_2 + \|\frac{f_3(x_1, t)}{p_t(x_1)} - \frac{h_3(x_1, t)}{p_t(x_1)}\|_2)$$
$$\leq C_1 \cdot |\alpha_t''| \cdot (C_5 \cdot \sqrt{\log N}|p_t(x_1) - f_1(x_1, t)| + \|f_2(x_1, t) - h_2(x_1, t)\|_2)$$

$$+ C_1 \cdot |\beta_t''| \cdot (C_5 \cdot |p_t(x_1) - f_1(x_1, t)| + \|f_3(x_1, t) - h_3(x_1, t)\|_2)$$

$$\leq \widetilde{C} \cdot ((|\alpha_t''| \cdot \sqrt{\log N} + |\beta_t''|) \cdot |f_1(x_1, t) - p_t(x_1)| + |\alpha_t''| \cdot \|f_2(x_1, t) - h_2(x_1, t)\|_2$$

$$+ |\beta_t''| \cdot \|f_3(x_1, t) - h_3(x_1, t)\|_2)$$

where the first step follows from Triangle Inequality, the second step follows from $|\alpha_t'' \leq \beta_t''|$, the third step follows from Lemma 5.20 and $\|\frac{f_2(x_1,t)}{f_1(x_1,t)}\|_2 \leq C_5\sqrt{\log N}$ and $\|\frac{f_3(x_1,t)}{f_1(x_1,t)}\|_2 \leq C_5$, and the last step follows from combining constants.

Further, we define $I_{B,1}$, with respect to **Part 1.**, as follows

$$I_{B,1} := \int_{\|x_1\|_\infty \leq \beta_t} \mathbf{1}[p_t(x_1) \geq N^{-\frac{2s+\omega}{d}}] \cdot \|f_4(x_1, t) - a_t(x_1)\|_2^2 \cdot p_t(x_1)\mathrm{d}x_1$$

$$\leq C' \cdot ((\alpha_t''^2 \log N + \beta_t''^2)$$

$$\cdot \underbrace{\int_{\|x_1\|_\infty \leq \beta_t} \mathbf{1}[p_t(x_1) \geq N^{-\frac{2s+\omega}{d}}] \cdot \|f_1(x_1, t) - p_t(x_1)\|_2^2 \cdot p_t(x_1)\mathrm{d}x_1}_{:=J_{B,1}}$$

$$+ \alpha_t''^2 \cdot \underbrace{\int_{\|x_1\|_\infty \leq \beta_t} \mathbf{1}[p_t(x_1) \geq N^{-\frac{2s+\omega}{d}}] \cdot \|f_2(x_1, t) - h_2(x_1, t)\|_2^2 \cdot p_t(x_1)\mathrm{d}x_1}_{:=J_{B,2}}$$

$$+ \beta_t''^2 \cdot \underbrace{\int_{\|x_1\|_\infty \leq \beta_t} \mathbf{1}[p_t(x_1) \geq N^{-\frac{2s+\omega}{d}}] \cdot \|f_3(x_1, t) - h_3(x_1, t)\|_2^2 \cdot p_t(x_1)\mathrm{d}x_1)}_{:=J_{B,3}}.$$

By definition of $f_2$ and $h_2$, we have

$$f_2(x_1, t) - h_2(x_1, t) = \int_{I^d} \frac{x_1 - \beta_t y}{\alpha_t} \cdot \frac{1}{(\sqrt{2\pi}\alpha_t)} \cdot \exp(-\frac{\|x_1 - \beta_t y\|_2^2}{2\alpha_t^2}) \cdot (f_N(y) - p_0(y))\mathrm{d}y.$$

Here, we only bound $J_{B,2}$ as an example:

$$J_{B,2} \leq C_1 \cdot \int_{\|x_1\|_\infty \leq \beta_t} \mathbf{1}[p_t(x_1) \geq N^{-\frac{2s+\omega}{d}}] \cdot \| \int_{f^d} \frac{x_1 - \beta_t y}{\alpha_t} \cdot \frac{1}{(\sqrt{2\pi}\alpha_t)^d}$$

$$\cdot \exp(-\frac{\|x_1 - \beta_t y\|_2^2}{2\alpha_t^2}) \cdot (f_N(y) - p_0(y))\mathrm{d}y\|_2^2\mathrm{d}x_1$$

$$\leq C_1 \cdot \int_{\|x_1\|_\infty \leq \beta_t} \|\frac{1}{\beta_t^d} \int_{\mathbb{R}^d} \mathbf{1}[\|y\|_\infty \leq 1] \cdot \frac{x_1 - \beta_t y}{\alpha_t} \cdot (\frac{\beta_t}{\sqrt{2\pi}\alpha_t})^d$$

$$\cdot \exp(-\frac{\|y - x_1/\beta_t\|_2^2}{2(\alpha_t/\beta_t)^2}) \cdot (f_N(y) - p_0(y))\mathrm{d}y\|_2^2\mathrm{d}x_1$$

$$\leq \frac{C_1}{\beta_t^2 d} \cdot \int_{\|x_1\|_\infty \leq \beta_t} \int_{\mathbb{R}^d} \mathbf{1}[\|y\|_\infty \leq 1] \cdot \|\frac{x_1 - \beta_t y}{\alpha_t}\|_2^2 \cdot (\frac{\beta_t}{\sqrt{2\pi}\alpha_t})^d$$

$$\cdot \exp(-\frac{\|y - x_1/\beta_t\|_2^2}{2(\alpha_t/\beta_t)^2}) \cdot (f_N(y) - p_0(y))^2\mathrm{d}y\mathrm{d}x_1$$

$$= \frac{C_1}{\beta_t^d} \cdot \int_{\|x_1\|_\infty \leq \beta_t} \int_{\mathbb{R}^d} \mathbf{1}[\|y\|_\infty \leq 1] \cdot \|\frac{x_1 - \beta_t y}{\alpha_t}\|_2^2 \cdot \frac{1}{(\sqrt{2\pi}\alpha_t)^d}$$

$$\cdot \exp(-\frac{\|x_1 - \beta_t y\|_2^2}{2\alpha_t^2}) \cdot (f_N(y) - p_0(y))^2\mathrm{d}y\mathrm{d}x_1,$$

where the third step follows from Jensen's inequality.

For $t \in 3N^{-\frac{\kappa^{-1}-\delta}{d}}$ with sufficiently large $N$, we can find $c_0 > 0$ such that $m_t \geq c_0$ on the time interval $[T_0, 3N^{-\frac{\kappa^{-1}-\delta}{d}}]$. We can thus further obtain for some $C' > 0$

$$J_{B,2} \leq C' \cdot \int_{I^d} \int_{\mathbb{R}^d} \cdot \frac{|x_1 - \beta_t y_l|^2}{\alpha_t^2} \cdot \frac{1}{(\sqrt{2\pi}\alpha_t)^d} \cdot \exp(-\frac{\|x_1 - \beta_t y_l\|_2^2}{2\alpha_t^2})\mathrm{d}x_1(f_N(y) - p_0(y))^2\mathrm{d}y$$

$$= dC' \cdot \int_{I^d} (f_N(y) - p_0(y))^2 \mathrm{d}y$$

$$= dC' \cdot \|f_N - p_0\|_{L^2(L^d)}$$

$$\leq C'' \cdot N^{\frac{-2s}{d}}$$

Similar to the above bound, we can prove that $J_{B,1}, J_{B,3}$ have the same bound of $N^{-\frac{2s}{d}}$ order. Thus there exist $C_{B_1} > 0$ such that

$$I_{B,1} \leq C_{B,1} \cdot (\alpha_t''^2 \log N + \beta_t''^2) \cdot N^{-\frac{2s}{d}}. \tag{4}$$

**Proof of Part 2.** $\beta_t \leq \|x_1\|_\infty \leq \beta_t + C_4$

Resort the bound $p_t(x_1) \geq N^{-(2s+\omega)/d}$, we have $1/p_t(x_1) \leq N^{(2s+\omega)/d}$, further we assume $1/f(x_1, t) \leq N^{(2s+\omega)/d}$. We have:

$$\|\frac{\alpha_t'' f_2(x_1, t) + \beta_t'' f_3(x_1, t)}{f_1(x_1, t)} - \frac{\alpha_t'' h_2(x_1, t) + \beta_t'' h_3(x_1, t)}{p_t(x_1)}\|_2$$

$$\leq \frac{1}{f_1(x_1, t)} \cdot \|(\alpha_t'' f_2(x_1, t) + \beta_t'' f_2(x_1, t)) - (\alpha_t'' h_2(x_1, t) + \beta_t'' h_3(x_1, t))\|_2$$

$$+ \|v_t(x_1)\|_2 \cdot \frac{1}{f_1(x_1, t)} \cdot |f_1(x_1, t) - p_t(x_1)|$$

$$\leq N^{(2s+\omega)/d} \cdot C \cdot ((\alpha_t''^2 \log N + \beta_t''^2) \cdot |p_t(x_1) - f_1(x_1, t)| + |\alpha_t''| \cdot |f_2(x_1, t) - h_2(x_1, t)|$$

$$+ |\beta_t''| \cdot |f_3(x_1, t) - h_3(x_1, t)|),$$

where the last step follows from $1/p_t(x_1) \leq N^{(2s+\omega)/d}$ and $1/f(x_1, t) \leq N^{(2s+\omega)/d}$.

We define $\Delta_{t,N} := \{x_1 \in \mathbb{R}^d \mid \beta_t \leq \|x_1\|_\infty \leq \beta_t + C_4 \alpha_t \sqrt{\log N}\}$ as the integral region and we have

$$I_{B,2} := \int_{\Delta_{t,N}} \mathbf{1}[p_t(x_1) \geq N^{-\frac{2s+\omega}{d}}] \cdot \|f_4(x_1, t) - v_t(x_1)\|_2^2 \cdot p_t(x_1) \mathrm{d}x_1$$

$$\leq C''' N^{\frac{4s+2\omega}{d}} \cdot (\alpha_t''^2 \log N + \beta_t''^2)$$

$$\cdot \underbrace{\int_{\Delta_{t,N}} \int_{I^d} \frac{1}{(\sqrt{2\pi}\alpha_t)^d} \cdot \exp(-\frac{\|x_1 - \beta_t y\|_2^2}{2\alpha_t^2} \cdot (f_N(y) - p_0(y))^2 \mathrm{d}y \mathrm{d}x_1)}_{:=K_{B,1}}$$

$$+ \alpha_t''^2$$

$$\cdot \underbrace{\int_{\Delta_{t,N}} \int_{I^d} \|\frac{x_1 - \beta_t y}{\alpha_t}\|_2^2 \cdot \frac{1}{(\sqrt{2\pi}\alpha_t)^d} \cdot \exp(-\frac{\|x_1 - \beta_t y\|_2^2}{2\alpha_t^2} \cdot (f_N(y) - p_0(y))^2 \mathrm{d}y \mathrm{d}x_1)}_{:=K_{B,2}}$$

$$+ \beta_t''^2 \cdot \underbrace{\int_{\Delta_{t,N}} \int_{I^d} \|y\|_2^2 \cdot \frac{1}{(\sqrt{2\pi}\alpha_t)^d} \cdot \exp(-\frac{\|x_1 - \beta_t y\|_2^2}{2\alpha_t^2} \cdot (f_N(y) - p_0(y))^2 \mathrm{d}y \mathrm{d}x_1)}_{:=K_{B,3}}.$$

Because of the factor $N^{(4s+2\omega)/d}$, the integrals need to be handled with lower orders compared to $J_{B,2}$ in order to obtain the desired bound on $I_{B,2}$. We utilize Assumption 5.2 regarding the higher-order smoothness near the boundary of $I^d$.

Since the three integrals admit analogous bounding strategies, we only examine the second integral, denoted by $K_{B,2}$. Given that $\delta, \omega > 0$ can be arbitrarily small, we impose $\check{s} > 6s + \delta\kappa + 2\omega$. Given range $A_{x_1} := \{y \in I^d \mid \|y - \frac{x_1}{\beta_t}\|_\infty \leq C_b \alpha_t \sqrt{\log N}/\beta_t\}$, we have

$$|\int_{I^d} \|\frac{x_1 - \beta_t y}{\alpha_t}\|_2^2 \cdot \frac{1}{(\sqrt{2\pi}\alpha_t)^d} \cdot \exp(-\frac{\|x_1 - \beta_t y\|_2^2}{2\alpha_t^2}) \cdot (f_N(y) - p_0(y))^2 \mathrm{d}y$$

$$-\int_{A_{x_1}} \|\frac{x_1 - \beta_t y}{\alpha_t}\|_2^2 \cdot \frac{1}{(\sqrt{2\pi}\alpha_t)^d} \cdot \exp(-\frac{\|x_1 - \beta_t y\|_2^2}{2\alpha_t^2}) \cdot (f_N(y) - p_0(y))^2 dy| \leq N^{-\frac{\check{s}}{d}}$$

which follows from Lemma 5.21 with $\epsilon = N^{-\check{s}/d}$.

Note that if $x_1 \in \Delta_{t,N}$ and $y \in A_{x_1}$, then

$$-1 \leq y_j \leq -1 + C_b \alpha_t \sqrt{\log N}/\beta_t$$

or

$$1 - C_b \alpha_t \sqrt{\log N}/\beta_t \leq y_j \leq 1$$

holds for each $j \in [d]$.

Since we assume $t \leq 3N^{-\frac{\kappa^{-1}-\delta}{d}}$ and $\alpha_t = b_0 t^\kappa$, we can deduce from Assumption 5.4 that $\beta_t \geq \sqrt{D_0}/2$. For sufficiently large $N$, this ensures $y$ is in the space $y \in I^d \setminus I_N^d$. We can show that:

$$\frac{C_b \alpha_t \sqrt{\log N}}{\beta_t} \leq \frac{2C_b b_0 3^\kappa}{\sqrt{D_0}} N^{-\frac{1-\delta\kappa}{d}},$$

which follows from $\alpha_t = b_0 t^\kappa \leq b_0 3^\kappa N^{-\frac{1-\delta\kappa}{d}}$.

Then we have

$$K_{B,2} = \int_{\Delta_{t,N}} \int_{I^d} \|\frac{x_1 - \beta_t y}{\alpha_t^2}\|_2^2 \cdot \frac{1}{(\sqrt{2\pi}\alpha_t)^d} \cdot \exp(-\frac{\|x_1 - \beta_t y\|_2^2}{2\alpha_t^2})$$
$$\cdot (f_N(y) - p_0(y))^2 dy dx_1$$
$$\leq \int_{\Delta_{t,N}} (\int_{A_{x_1}} \|\frac{x_1 - \beta_t y}{\alpha_t^2}\|_2^2 \cdot \frac{1}{(\sqrt{2\pi}\alpha_t)^d} \cdot \exp(-\frac{\|x_1 - \beta_t y\|_2^2}{2\alpha_t^2})$$
$$\cdot (f_N(y) - p_0(y))^2 dy + N^{-\check{s}/d}) dx_1$$
$$\leq \int_{I^d \setminus I_N^d} \int_{\mathbb{R}^d} \|\frac{x_1 - \beta_t y}{\alpha_t^2}\|_2^2 \cdot \frac{1}{(\sqrt{2\pi}\alpha_t)^d} \cdot \exp(-\frac{\|x_1 - \beta_t y\|_2^2}{2\alpha_t^2}) dx_1$$
$$\cdot (f_N(y) - p_0(y))^2 dy + N^{-\check{s}/d} |\Delta_{t,N}|.$$

which follows from $\|f_N - p_0\|_{L^2(I^d \setminus I_N^d)} \leq N^{-\check{s}/d}$ holds for $y \in I^d \setminus I_N^d$.

Since the volume $|\Delta_{t,N}|$ is upper bounded by $D'\alpha_t \sqrt{\log N}$ with some constant $D' > 0$, we have

$$K_{B,2} \leq d\|f_N - p_0\|_{L^2(I^d \setminus I_N^d)}^2 + C' \cdot (\alpha_t \sqrt{\log N}) \cdot N^{-\check{s}/d}$$
$$\leq C'' \cdot (N^{-2\check{s}/d} + N^{-(\check{s}+1-\delta\kappa)/d} \cdot \log^{d/2} N)$$
$$= O(N^{-(\check{s}+1-\delta\kappa)/d} \cdot \log^{d/2} N),$$

where the last step follows from Assumption 5.2.

$K_{B,1}$ and $K_{B,3}$ share the similar bound of $K_{B,2}$. Consequently, there is a constant $C_{B,2} > 0$, independent of $n$ and $t$, for which

$$I_{B,2} \leq C_{B,2} \cdot (\alpha_t''^2 \log N + \beta_t''^2) \cdot N^{-(\check{s}+1-4s-2\omega-\delta\kappa)/d}.$$

Since we have taken $\check{s}$ so that $\check{s} > 6s + \delta\kappa + 2\omega$, we have

$$I_{B,2} \leq C_{B,2} \cdot (\alpha_t''^2 \log N + \beta_t''^2) \cdot N^{-2s/d}. \tag{5}$$

It follows from Eq. (4) and Eq. (5) that there is $C_B > 0$ such that

$$I_B \leq C_B \cdot (\alpha_t''^2 \log N + \beta_t''^2) \cdot N^{-2s/d}.$$

**Conclusion.**

combine **Part 1.** and **Part 2.**, we can bound $I_B$. Combine the bound of $I_A$ and $I_B$, we prove this theorem.

$$\square$$

## E  Bounds on Second Order Flow Matching for Large $t$

We formally present the proof of Theorem 4.2 in this section.

**Theorem E.1** (Main Theorem, Bound Acceleration Error under Large $t$, Formal Version of Theorem 4.2)**.** *If the following conditions hold:*

- *Fix $t_* \in [T_*, 1]$ and take arbitrary $\eta > 0$.*

- *Assume Assumption 5.2, 5.3, 5.4, 5.6, 5.8, 5.9 hold.*

- *Let $C_7$ be a constant independent of $t$.*

- *Let $x_1$ be the trajectory, $x_2 := \phi_2(x_1, t)$ where $\phi_2$ is the neural network in Lemma B.4.*

- *Let $x$ be defined as the concatenation of $x_1$ and $x_2$, i.e., $x := [x_1, x_2]$.*

*Then there exists a neural network $u_2 \in \mathcal{M}(L, W, S, B)$ and a constant $C_7 > 0$, independent of $t$, such that the bound*

$$\int \|u_2(x, t) - a_t(x_1)\|_2^2 \cdot p_t(x_1) \mathrm{d}x_1 \leq C_7 \cdot (\alpha_t''^2 \log N + \beta_t''^2) \cdot N^{-\eta}$$

*holds for all $t \in [2t_*, 1]$, where*

$$L = O(\log^4 N), \|W\|_\infty = O(N \log^6 N), S = O(N \log^8 N), B = \exp(O((\log N) \cdot (\log \log N))).$$

*Proof.* For any $t \in [N^{-(\kappa^{-1} - \delta)/d}, 1]$ and some constant $C_8$ independent of $t$, we can show that

$$\int_{\|x_1\|_2 \geq \beta_t + C_8 \sqrt{\log N}} p_t(x_1) \cdot \|u_2(x, t) - a_t(x_1)\|_2^2 \mathrm{d}x_1 \lesssim (|\alpha_t''| \sqrt{\log N} + |\beta_t''|) \cdot N^{-\eta}$$

which follows from $\|u_2(x, t)\|_2 \leq C_3 \cdot (|\alpha_t''| \sqrt{\log N} + |\beta_t''|)$.

We can show that

$$\int_{\mathbb{R}^d} p_t(x_1) \cdot \|u_2(x, t) - a_t(x_1)\|_2^2 \mathrm{d}x_1$$

$$= \int_{\|x_1\|_2 \leq \beta_t + C_8 \sqrt{\log N}} \mathbf{1}[p_t(x_1) \geq N^{-\eta}] \cdot p_t(x_1) \cdot \|u_2(x, t) - a_t(x_1)\|_2^2 \mathrm{d}x_1$$

$$+ O((|\alpha_t''| \sqrt{\log N} + |\beta_t''|) \cdot N^{-\eta}).$$

which follow from similar bound as Lemma 5.13.

Thus we can focus on the integral region $\{x_1 \mid p_t(x_1) \geq N^{-\eta}\}$. We use $B$-spline approximation on $p_t(x_1)$. we rewrite $p_t(x_1)$ as follows:

$$p_t(x_1) = \int_{\mathbb{R}^d} \frac{1}{(\sqrt{2\pi}\widetilde{\alpha}_t)^d} \cdot \exp(-\frac{\|x_1 - \widetilde{\beta}_t y\|_2^2}{2\widetilde{\alpha}_t^2}) \cdot p_{t_*}(y) \mathrm{d}y,$$

where

$$\widetilde{\beta}_t := \frac{\beta_t}{\beta_{t_*}}, \ \widetilde{\alpha}_t := \sqrt{\alpha_t^2 - (\frac{\beta_t}{\beta_{t_*}})^2 \cdot \alpha_{t_*}^2}.$$

We can thus apply a similar argument to Theorem D.1.

We use a $B$-spline approximation of $p_{t_*}$. For $\eta > 0$, take $\mathcal{A} \in \mathbb{N}$ such that $\mathcal{A} > \frac{3d\eta}{2\delta\kappa}$. We can show that for any $k \leq \mathcal{A}$ and any tuple $(i_1, \ldots, i_k)$

$$\|\frac{\mathrm{d}^k p_{t*}(x_1)}{\mathrm{d}x_{1,i_1} \cdots \mathrm{d}x_{1,i_k}}\|_2 \leq \frac{C_a}{\alpha_{t_*}^k},$$

which follows from Lemma 5.16.

There exist $t_{0,*} \in [0,1]$ and $b_{0,*} > 0$ such that

$$\alpha_t \geq b_{0,*} t^\kappa$$

holds for any $0 \leq t \leq t_{0,*}$, which follows from Assumption 5.4. Let's define $c_{0,*} := (t_{0,*})^\kappa > 0$, thus for any $t \in [0,1]$, we have

$$\alpha_t \geq \max\{b_{0,*} t^\kappa, c_{0,*}\}.$$

We can show that

$$\frac{p_{t_*}}{\max\{t_*^{-\mathcal{A}\kappa}, c_{0,*}\}} \in B_{\infty,\infty}^{\mathcal{A}}(\mathbb{R}^d)$$

holds, because for any $k \leq \mathcal{A}$ we have

$$
\begin{aligned}
\|\frac{\mathrm{d}^k}{\mathrm{d}x_{1,i_1} \cdots \mathrm{d}x_{1,i_k}} \cdot \frac{p_{t_*}(x_1)}{\max\{t_*^{-\mathcal{A}\kappa}, c_{0,*}\}}\|_2 &\leq \frac{C_\mathcal{A} \cdot \min\{b_{0,*} t_*^{-k\kappa}, (c_{0,*})^{-k}\}}{\max\{t_*^{-\mathcal{A}\kappa}, c_{0,*}\}} \\
&\leq C_\mathcal{A} \cdot \min\{b_{0,*} t_*^{(\mathcal{A}-k)\kappa}, (c_{0,*})^{-(k+1)}\} \\
&\leq C_\mathcal{A} \cdot \min\{b_{0,*}, (c_{0,*})^{-(k+1)}\},
\end{aligned}
$$

which implies $\frac{p_{t_*}}{\max\{t_*^{-\mathcal{A}\kappa}, c_{0,*}\}} \in W_\infty^\mathcal{A}(\mathbb{R}^d)$ and $\|\frac{p_{t_*}}{\max\{t_*^{-\mathcal{A}\kappa}, c_{0,*}\}}\|_{W_\infty^\mathcal{A}(\mathbb{R}^d)} \leq C_\mathcal{A} \cdot \min\{b_{0,*}, (c_{0,*})^{-(k+1)}\}$ for some constant $C_\mathcal{A}$.

There is $C_5 > 0$ such that

$$\int_{\|y\|_\infty \geq C_5 \sqrt{\log N}} (\|y\|_2^2 + 1) \cdot p_{t_*}(y) \mathrm{d}y \leq N^{-3\eta} \tag{6}$$

which follows from that we use a similar argument as in the proof of Lemma 5.15.

Considering a $B$-spline approximation on $[-C_5\sqrt{\log N}, C_5\sqrt{\log N}]^d$ we define

$$N_* := \lceil t_*^{-d\kappa} \cdot N^{\delta\kappa} \rceil$$

be the number of $B$-spline bases. By Theorem 5.24, we can define function $f_{N^*}$ of the form

$$f_{N^*}(x_1) = \max\{t_*^{-\mathcal{A}\kappa}, c_{0,*}\} \cdot \sum_{i=1}^{N^*} \mathcal{A}_i \cdot \mathbf{1}[\|x_1\|_\infty \leq C_5 \sqrt{\log N}] \cdot M_{k_i,j_i}^d(x_1)$$

with $|\mathcal{A}_i| \leq 1$ and $C_9 > 0$ such that

$$\|p_{t_*} - f_{N^*}\|_{L^2([-C_5\sqrt{\log N}, C_5\sqrt{\log N}]^d)} \leq C_9 \cdot (\log N)^{\mathcal{A}/2} \cdot (N^*)^{-\frac{\mathcal{A}}{d}} \cdot \max\{t_*^{-\mathcal{A}\kappa}, c_{0,*}\}$$

holds.

For sufficiently large $N$, we can show that

$$C' \cdot (\log N)^{\mathcal{A}/2} \cdot N^{-\delta\mathcal{A}\kappa/d} \leq C' \cdot N^{-3\eta/2}$$

which follows from $N^* \geq t_*^{-d\kappa} \cdot N^{\delta\kappa}$ and $\mathcal{A} > \frac{3d\eta}{2\delta\kappa}$. Above bound implies that

$$\|p_{t_*} - f_{N^*}\|_{L^2([-C_5\sqrt{\log N}, C_5\sqrt{\log N}]^d)} \leq C_{10} \cdot N^{-3\eta/2} \tag{7}$$

holds for sufficiently large $N$ and some constant $C_{10}$. Thus we have

$$|\widetilde{\mathcal{A}}_i| \leq \max\{t_*^{-\mathcal{A}\kappa}, c_{0,*}\} \leq N^{(\kappa^{-1}-\delta)/d\mathcal{A}\kappa} = N^{\mathcal{A}(1-\delta\kappa)/d}.$$

Similar to proof of Theorem D.1, we define $f_1$ using $f_{N^*}$ as

$$f_1(x_1, t) := \max\{\widetilde{f}_1(x_1, t), N^{-\eta}\},$$

where we define $\widetilde{f}_1(x_1, t)$ as

$$\widetilde{f}_1(x_1, t) := \int \frac{1}{(\sqrt{2\pi}\widetilde{\alpha}_t)^d} \cdot \exp(-\frac{\|x_1 - \widetilde{\beta}_t y\|_2^2}{2\widetilde{\alpha}_t^2}) \cdot f_{N^*}(y)\mathrm{d}y.$$

We also define $f_2$, $f_3$, and $f_4$ as

$$f_2(x_1, t) := \int \frac{x_1 - \widetilde{\beta}_t y}{\widetilde{\alpha}_t} \cdot \frac{1}{(\sqrt{2\pi}\widetilde{\alpha}_t)^d} \cdot \exp(-\frac{\|x_1 - \widetilde{\beta}_t y\|_2^2}{2\widetilde{\alpha}_t^2}) \cdot f_{N^*}(y)\mathrm{d}y,$$

$$f_3(x_1, t) := \int y \cdot \frac{1}{(\sqrt{2\pi}\widetilde{\alpha}_t)^d} \cdot \exp(-\frac{\|x_1 - \widetilde{\beta}_t y\|_2^2}{2\widetilde{\alpha}_t^2}) \cdot f_{N^*}(y)\mathrm{d}y,$$

$$f_4(x_1, t) := \frac{\widetilde{\alpha}_t'' f_2(x_1, t) + \widetilde{\beta}_t'' f_3(x_1, t)}{f_1(x_1, t)} \cdot \mathbf{1}[|\frac{f_2(x_1, t)}{f_1(x_1, t)}| \leq C_5 \cdot \sqrt{\log N}] \cdot \mathbf{1}[|\frac{f_3(x_1, t)}{f_1(x_1, t)}| \leq C_5],$$

Similar to Eq. (3), we can bound $\widetilde{I}_A$ and $\widetilde{I}_B$ separately:

$$\int_D \mathbf{1}[p_t(x_1) \geq N^{-\eta}] \cdot \|u_2(x, t) - a_t(x_1)\|_2^2 \cdot p_t(x_1)\mathrm{d}x_1$$

$$\leq \int_D \mathbf{1}[p_t(x_1) \geq N^{-\eta}] \cdot \|u_2(x, t) - f_4(x_1, t)\|_2^2 \cdot p_t(x_1)\mathrm{d}x_1$$

$$+ \int_D \mathbf{1}[p_t(x_1) \geq N^{-\eta}] \cdot \|f_4(x_1, t) - a_t(x_1)\|_2^2 \cdot p_t(x_1)\mathrm{d}x_1$$

$$=: \widetilde{I}_A + \widetilde{I}_B.$$

**Bound of $\widetilde{I}_A$.**

Using the same step as **Bound of $I_A$** in the proof of Theorem D.1, we can show

$$\widetilde{I}_A = O(\mathrm{poly}(\log N) \cdot N^{-\eta'})$$

for arbitrary $\eta' > 0$ such that we can omit it.

**Bound of $\widetilde{I}_B$.**

we define $h_2(x_1, t)$ and $h_3(x_1, t)$ as follows

$$h_2(x_1, t) := \int_{\mathbb{R}^d} \frac{x_1 - \widetilde{\beta}_t y}{\widetilde{\alpha}_t} \cdot \frac{1}{(\sqrt{2\pi}\widetilde{\alpha}_t)^d} \cdot \exp(-\frac{\|x_1 - \widetilde{\beta}_t y\|_2^2}{2\widetilde{\alpha}_t^2}) \cdot p_{t_*}(y)\mathrm{d}y$$

$$h_3(x_1, t) := \int_{\mathbb{R}^d} y \cdot \frac{1}{(\sqrt{2\pi}\widetilde{\alpha}_t)^d} \cdot \exp(-\frac{\|x_1 - \widetilde{\beta}_t y\|_2^2}{2\widetilde{\alpha}_t^2}) \cdot p_{t_*}(y)\mathrm{d}y.$$

Then we can show that

$$\|f_4(x_1, t) - a_t(x_1)\|_2 \leq N^\eta \widetilde{C} \cdot ((|\widetilde{\alpha}_t''|\sqrt{\log N} + |\widetilde{\beta}_t''|) \cdot |p_t(x_1) - f_1(x_1, t)|$$
$$+ |\widetilde{\alpha}_t''| \cdot \|f_2(x_1, t) - h_2(x_1, t)\|_2 + |\widetilde{\beta}_t''| \cdot \|f_3(x_1, t) - h_3(x_1, t)\|_2)$$

for some constant $\widetilde{C}$, and thus

$$\widetilde{I}_B \leq C' N^{2\eta} \cdot (((\widetilde{\alpha}_t'')^2 \log N + (\widetilde{\beta}_t'')^2)$$

$$\cdot \underbrace{\int_D |\int_{\mathbb{R}^d} \frac{1}{(\sqrt{2\pi}\widetilde{\alpha}_t)^d} \cdot \exp(-\frac{\|x_1 - \widetilde{\beta}_t y\|_2^2}{2\widetilde{\alpha}_t^2} \cdot (f_{N^*}(y) - p_{t_*}(y))\mathrm{d}y|^2 \mathrm{d}x_1)}_{\widetilde{J}_{B,1}}$$

$$+ (\widetilde{\alpha}_t'')^2$$

$$\cdot \underbrace{\int_D \|\int_{\mathbb{R}^d} \frac{x_1 - \widetilde{\beta}_t y}{\widetilde{\alpha}_t} \cdot \frac{1}{(\sqrt{2\pi}\widetilde{\alpha}_t)^d} \cdot \exp(-\frac{\|x_1 - \widetilde{\beta}_t y\|_2^2}{2\widetilde{\alpha}_t^2} \cdot (f_{N^*}(y) - p_{t_*}(y))\mathrm{d}y\|_2^2 \mathrm{d}x_1)}_{:=\widetilde{J}_{B,2}}$$

$$+ (\widetilde{\beta}''_t)^2 \cdot \underbrace{\int_D \| \int_{\mathbb{R}^d} y \frac{1}{(\sqrt{2\pi}\widetilde{\alpha}_t)^d} \cdot \exp(-\frac{\|x_1 - \widetilde{\beta}_t y\|_2^2}{2\widetilde{\alpha}_t^2} \cdot (f_{N^*}(y) - p_{t_*}(y)) \mathrm{d}y \|_2^2 \mathrm{d}x_1)}_{\widetilde{J}_{B,3}}).$$

In the following part, our goal is to bound $\widetilde{J}_{B,2}$. By employing the same method, we can also handle $\widetilde{J}_{B,1}$ and $\widetilde{J}_{B,3}$ similarly.

We introduce $\rho := \frac{1}{\sqrt{2}D_0} > 0$, where $D_0$ is given in Assumption 5.4. We then bound $\widetilde{J}_{B,2}$ across different region of $t$.

- **Part 1.** $\widetilde{\beta}_t \geq \rho$.

- **Part 2.** $\widetilde{\beta}_t \leq \rho$.

Let's prove step by step.

**Proof of Part 1.** $\widetilde{\beta}_t \geq \rho$.

By rewriting the inner integral on $y$ by a Gaussian integral, we have

$$\widetilde{J}_{B,2} = \int_D \frac{1}{\widetilde{\beta}_t^{2d}} \cdot \| \int_{\mathbb{R}^d} \frac{x_1 - \widetilde{\beta}_t y}{\widetilde{\alpha}_t} \cdot (\frac{\widetilde{\beta}_t}{\sqrt{2\pi}\widetilde{\alpha}_t})^d \cdot \exp(-\frac{\widetilde{\beta}_t^2 \|y - x_1/\widetilde{\beta}_t\|_2^2}{2\widetilde{\alpha}_t^2})$$
$$\cdot (f_{N^*}(y) - p_{t_*}(y)) \mathrm{d}y \|_2^2 \mathrm{d}x_1$$
$$\leq \int_D \frac{1}{\widetilde{\beta}_t^{2d}} \cdot \int_{\mathbb{R}^d} \| \frac{x_1 - \widetilde{\beta}_t y}{\widetilde{\alpha}_t} \|_2^2 \cdot (\frac{\widetilde{\beta}_t}{\sqrt{2\pi}\widetilde{\alpha}_t})^d \cdot \exp(-\frac{\widetilde{\beta}_t^2 \|y - x_1/\widetilde{\beta}_t\|_2^2}{2\widetilde{\alpha}_t^2})$$
$$\cdot (f_{N^*}(y) - p_{t_*}(y))^2 \mathrm{d}y \mathrm{d}x_1$$
$$\leq \int_D \frac{1}{\widetilde{\beta}_t^d} \cdot \int_{\mathbb{R}^d} \| \frac{x_1 - \widetilde{\beta}_t y}{\widetilde{\alpha}_t} \|_2^2 \cdot \frac{1}{(\sqrt{2\pi}\widetilde{\alpha}_t)^d} \cdot \exp(-\frac{\|x_1 - \widetilde{\beta}_t y\|_2^2}{2\widetilde{\alpha}_t^2})$$
$$\cdot (f_{N^*}(y) - p_{t_*}(y))^2 \mathrm{d}y \mathrm{d}x_1$$
$$\leq (2D_0)^{d/2} \cdot \int_{\mathbb{R}^d} \int_{\mathbb{R}^d} \| \frac{x_1 - \widetilde{\beta}_t y}{\widetilde{\alpha}_t} \|_2^2 \cdot \frac{1}{(\sqrt{2\pi}\widetilde{\alpha}_t)^d} \cdot \exp(-\frac{\|x_1 - \widetilde{\beta}_t y\|_2^2}{2\widetilde{\alpha}_t^2})$$
$$\cdot (f_{N^*}(y) - p_{t_*}(y))^2 \mathrm{d}x_1 \mathrm{d}y$$
$$\leq \rho^{-d/2} d \cdot \int_{\mathbb{R}^d} (f_{N^*}(y) - p_{t_*}(y))^2 \mathrm{d}y$$
$$\leq \rho^{-d/2} d \cdot (\int_{[-C_5\sqrt{\log N}, C_5\sqrt{\log N}]^d} (f_{N^*}(y) - p_{t_*}(y))^2 \mathrm{d}y + \int_{\|y\|_2 \geq C_5\sqrt{\log N}} p_{t_*}(y)^2 \mathrm{d}y)$$
$$\leq \rho^{-d/2} d \cdot (\|f_{N^*} - p_{t_*}\|_{L^2([-C_5\sqrt{\log N}, C_5\sqrt{\log N}]^d)}^2 + N^{-3\eta})$$
$$\leq C \cdot N^{-3\eta}$$

where the second step follows from Jensen's inequality, the seventh step follows from Eq. (6), and the last step follows from Eq. (7).

**Proof of Part 2.** $\widetilde{\beta}_t \leq \rho$.

We have
$$\widetilde{\alpha}_t^2 = \alpha_t^2 - \widetilde{\beta}_t^2 \alpha_{t_*}^2 \geq \alpha_t^2 - \rho^2 \alpha_{t_*}^2. \tag{8}$$

which follows from $\widetilde{\beta}_t \leq \rho$.

Further, we can show that
$$\alpha_t^2 \geq D_0^{-1} - \beta_t^2 \geq D_0^{-1} - \rho^2 \beta_{t_*}^2. \tag{9}$$

which follows from Assumption 5.4 and $\beta_t^2 \le \rho^2 \beta_{t_*}^2$.

Further, we can show

$$\widetilde{\alpha}_t^2 \ge D_0^{-1} - \rho^2(\beta_{t_*}^2 + \alpha_{t_*}^2)$$
$$\ge D_0^{-1} - \rho^2 D_0$$
$$= \frac{1}{2D_0},$$

where the first step follows from Eq. (8) and Eq. (9), and the last step follows from the definition $\rho = \frac{1}{\sqrt{2D_0}}$.

We divide the integral of $\widetilde{J}_{B,2}$ into 2 regions: $\{y \mid \|y\|_\infty \ge C_5\sqrt{\log N}\}$ and $\{y \mid \|y\|_\infty \le C_5\sqrt{\log N}\}$.

In the region $\{y \mid \|y\|_\infty \ge C_5\sqrt{\log N}\}$, we can show that

$$\|\int_{\|y\|_\infty \ge C_5\sqrt{\log N}} \frac{x_1 - \widetilde{\beta}_t y}{\widetilde{\alpha}_t} \cdot \frac{1}{(\sqrt{2\pi}\widetilde{\alpha}_t)^d} \cdot \exp(-\frac{\|x_1 - \widetilde{\beta}_t y\|_2^2}{2\widetilde{\alpha}_t^2}) \cdot (f_{N^*}(y) - p_{t_*}(y))\mathrm{d}y\|_2^2$$

$$\le (\frac{D_0}{\pi})^d (2D_0)^2 \cdot \int_{\|y\|_\infty \ge C_5\sqrt{\log N}} \|x_1 - \widetilde{\beta}_t y\|_2^2 \cdot (p_{t_*}(y))^2 \mathrm{d}y$$

$$\le C \cdot (\frac{D_0}{\pi})^d (2D_0)^2 \cdot \int_{\|y\|_\infty \ge C_5\sqrt{\log N}} (C'\log N + \rho^{-2}\|y\|_2^2) \cdot p_{t_*}(y)\mathrm{d}y$$

$$\le C'' \cdot N^{-3\eta} \cdot \log N,$$

where the first step follows from $\widetilde{\alpha}_t^2 \ge 1/(2D_0)$ and $f_{N^*}(y) = 0$, and the last step follows from Eq. (6) and $\|x_1\|_2^2 \le C'\log N$ holds for some constant $C'$ when $x_1 \in D$.

Then we can show:

$$\|\int_{\|y\|_\infty \le C_5\sqrt{\log N}} \frac{x_1 - \widetilde{\beta}_t y}{\widetilde{\alpha}_t} \cdot \frac{1}{(\sqrt{2\pi}\widetilde{\alpha}_t)^d} \cdot \exp(-\frac{\|x_1 - \widetilde{\beta}_t y\|_2^2}{2\widetilde{\alpha}_t^2}) \cdot (f_{N^*}(y) - p_{t_*}(y))\mathrm{d}y\|_2^2$$

$$\le D'^2 \cdot \log N \cdot (\frac{D_0}{\pi}) \int_{\|y\|_\infty \le C_5\sqrt{\log N}} \mathrm{d}y \cdot \int_{\|y\|_\infty \le C_5\sqrt{\log N}} (f_{N^*}(y) - p_{t_*}(y))^2 \mathrm{d}y$$

$$\le D'' \cdot \log^{\frac{d}{2}+1} N \cdot \|f_{N^*} - p_{t_*}\|_{L^2([-C_5\sqrt{\log N}^2, C_5\sqrt{\log N}]^d)}$$

$$\le D'' \cdot N^{-3\eta} \cdot \log^{\frac{d}{2}+1} N.$$

where the first step follows from $\|x_1 - \widetilde{\beta}_t y\|_2/\widetilde{\alpha}_t \le D'\sqrt{\log N}$ for some $D' > 0$ when $x_1 \in D$, the second step follows from Cauchy-Schwarz inequality.

Following from **Part 1.** and **Part 2.**, we can bound $\widetilde{J}_{B,2}$

$$\widetilde{J}_{B,2} \le \mathrm{poly}(\log N) \cdot N^{-\eta}.$$

Finally, we can show that there exist a constant $C''$ such that

$$\widetilde{I}_B \le C'' \cdot (\alpha_t''^2 \log N + \beta_t''^2) \cdot N^{-\eta} \cdot \mathrm{poly}(\log N).$$

where we can omit $\mathrm{poly}(\log N)$ when we use enough large $\eta$. $\qquad\square$

## LLM USAGE DISCLOSURE

LLMs were used only to polish language, such as grammar and wording. These models did not contribute to idea creation or writing, and the authors take full responsibility for this paper's content.

