# OpenReview forum: "Theoretical Guarantees for High Order Trajectory Refinement in Generative Flows"
_ICLR.cc/2026/Conference — Submitted to ICLR 2026_

### Official Review · Reviewer_pSP1 · 2025-10-29

**Soundness:** 3
**Presentation:** 3
**Contribution:** 1
**Rating:** 2
**Confidence:** 3

**Summary:**

This paper builds on the work by Oko et al and Fukumizu et al on the generalization bound of Diffusion process and Flow Matching. The paper brings to the attention that higher order matching methods are emerging, and show that the acceleration matching loss (an approximation risk)  can be bounded from above by the worst case optimal convergence rate.

**Strengths:**

The paper certainly has the credit of

**Weaknesses:**

I believe that the paper is weak in motivation. Fukumizu et al sets its goal in bounding the wasserstein error between the generated distribution and the target distribution, which is a direct, practical measure of the "wellness of genenaration".

It is very much probable that bounding this risk at the Besov-minimax rate guarantees that any consistent flow trained via this regression (acceleration matching) will yield distributions whose estimation error (in Wasserstein or TV) is of the same order — but the paper never formalizes that link.  Without such support, I am finding it hard to find the value of the results presented in this paper *on this conference venue*.

Also, it was to be hoped that the theoretical results would support the necessity of higher order matching in some way, but the paper does not seem to provide the connection.

**Questions:**

My greatest question is posed in the weakness section. Can the main results in this paper be directly related to a loss more practical then the the acceleration regression error  $E_{p_t}[ || u_1 - a ||]$ ?

**Details Of Ethics Concerns:**

Nothing in particular, this is a purely theoretical research.

---

### Official Review · Reviewer_HDXY · 2025-10-29

**Soundness:** 2
**Presentation:** 2
**Contribution:** 2
**Rating:** 4
**Confidence:** 2

**Summary:**

This paper presents an upper bound on the estimation error for second-order flow matching. They show that the error depends polynomially on the smoothness of the target distribution.

**Strengths:**

They present a bound for second-order flow matching, unlike previous works that only discuss first-order matching.

**Weaknesses:**

Is the use or training of second-order derivatives common in diffusion or flow-based literature? To my knowledge, their use is niche; most training-based acceleration approaches rely on explicitly learning integral (e.g., consistency models, reflow). Non-training-based approaches (e.g., better solvers) just use (and have to use) the first-order gradient. Clarifying the significance of the results would be helpful.

Additionally, the definition of  N is unclear. In the paper, N is introduced only in relation to other parameters and never explicitly defined. Because of this coupling, it is difficult to reason about how the bound behaves when, for instance, the neural network is fixed and the sample size increases, or vice versa.

> Furthermore, a multitude of studies have served as supplementary inspirations for our work (Xu et al., 2022a; Dax et al., 2023; Pooladian et al., 2023; Wang et al., 2023c;a; Shen et al., 2025a;b; Wang et al., 2024b; Chen & Lipman, 2024; Klein et al., 2024; Chen et al., 2025c; Cao et al., 2025; Cheng et al., 2024; Wang et al., 2023b; Feng et al., 2024b; Liu et al., 2024; Hu et al., 2024e).

Rather than listing a large block of references, it would be more informative to properly discuss these works and explain their connections to this study. Also, discussions of U-ViT, latent diffusion, or multi-scale noise schedules do not seem to directly clarify or contextualize the present work.

**Questions:**

Why not show the distributional bound between the true and generated distributions?

Does the main theorem provide non-trivial insights/guidelines for practitioners?

---

### Official Review · Reviewer_qiTf · 2025-10-31

**Soundness:** 2
**Presentation:** 2
**Contribution:** 2
**Rating:** 2
**Confidence:** 3

**Summary:**

The paper analyzes higher-order flow matching (with an acceleration term) and claims it preserves worst case optimality as a distribution estimator. It proves acceleration-error bounds for small-t (Theorem 4.1) and large-t (Theorem 4.2), with rates tied to Besov smoothness and neural-network complexity.

**Strengths:**

(1) To the best of my knowledge, this is the first paper to provide estimates within the higher-order Flow Matching framework, demonstrating nearly minimax-optimal convergence rates.
(2) Contributions and roadmap are clearly signposted.

**Weaknesses:**

(1) Within the framework of [1,2], the manuscript carries out analogous derivations for higher-order flow matching.
(2) Assumptions 5.2–5.9 and their role in the theoretical development are important; however, they appear to have already been clarified in [2].
(3) The essential estimates underlying the main results seem to occur in Lemmas 5.12–5.15. These lemmas primarily develop bounds for the higher-order components (e.g., acceleration terms), which, while technically sound, do not strike me as conceptually deep or revealing new insights beyond [2].

[1] Kazusato Oko, Shunta Akiyama, and Taiji Suzuki. Diffusion models are minimax optimal distribution estimators. volume 202, pages 26517–26582. PMLR, 4 2023
[2] Fukumizu K, Suzuki T, Isobe N, et al. Flow matching achieves almost minimax optimal convergence[J]. arXiv preprint arXiv:2405.20879, 2024.

**Questions:**

Could you clarify where the manuscript reveals a substantive, conceptually new contribution that departs from [2]?
Definition 3.10 mixes the symbols \alpha, \beta with a, b;
Assumptions 5.7–5.8 use the notation \kappa and \overline{\kappa}, but \overline{\kappa} is not defined earlier. Do you intend \overline{\kappa} to be the \tilde{\kappa} introduced in Assumption 5.4?
Assumptions 5.5–5.6 bound |\alpha''_t|+|\beta''_t|by K_0 N for all t \in [T_0,1], making these derivatives scale linearly in the sample-size parameter N. Could you justify this dependence?

---

### Meta-Review · Area_Chair_D6mV · 2025-12-09

**Summary:**

The paper analyzes higher-order flow matching and claims it preserves worst case optimality as a distribution estimator. Reviewers find many big issues. Author did not provide the rebuttal.

**Reviewer Concerns:**

Reviewers find many big issues. Author did not provide the rebuttal.

**Reviewer Scores:**

Reviewers find many big issues. Author did not provide the rebuttal.

---

### Decision · Program_Chairs · 2026-01-26

Reject